# TTVD: Towards a Geometric Framework for Test-Time Adaptation Based on Voronoi Diagram

**Mingxi Lei[1], Chunwei Ma[2], Meng Ding[1], Yufan Zhou[3], Ziyun Huang[4]\*, Jinhui Xu[5]†\***

[1]Department of Computer Science and Engineering, University at Buffalo, SUNY
[2]JMP Statistical Discovery LLC
[3]Adobe Research
[4]Computer Science and Software Engineering, Penn State Erie
[5]School of Information Science and Technology, University of Science and Technology of China
[1]{mingxile, mengding}@buffalo.edu, [2]chunwei.ma@jmp.com
[3]yufzhou@adobe.com, [4]zxh201@psu.edu, [5]jhxu00@gmail.com

## Abstract

Deep learning models often struggle with generalization when deploying on real-world data, due to the common distributional shift to the training data. Test-time adaptation (TTA) is an emerging scheme used at inference time to address this issue. In TTA, models are adapted online at the same time when making predictions to test data. Neighbor-based approaches have gained attention recently, where prototype embeddings provide location information to alleviate the feature shift between training and testing data. However, due to their inherit limitation of simplicity, they often struggle to learn useful patterns and encounter performance degradation. To confront this challenge, we study the TTA problem from a geometric point of view. We first reveal that the underlying structure of neighbor-based methods aligns with the Voronoi Diagram, a classical computational geometry model for space partitioning. Building on this observation, we propose the Test-Time adjustment by Voronoi Diagram guidance (TTVD), a novel framework that leverages the benefits of this geometric property. Specifically, we explore two key structures: **(I)** Cluster-induced Voronoi Diagram (CIVD): This integrates the joint contribution of self-supervision and entropy-based methods to provide richer information. **(II)** Power Diagram (PD): A generalized version of the Voronoi Diagram that refines partitions by assigning weights to each Voronoi cell. Our experiments under rigid, peer-reviewed settings on CIFAR-10-C, CIFAR-100-C, ImageNet-C, and ImageNet-R shows that TTVD achieves remarkable improvements compared to state-of-the-art methods. Moreover, extensive experimental results also explore the effects of batch size and class imbalance, which are two scenarios commonly encountered in real-world applications. These analyses further validate the robustness and adaptability of our proposed framework.

## 1 Introduction

Deep learning models have demonstrated impressive capabilities across a multitude of recognition tasks, thanks to substantial large datasets, advanced network architectures and computing capability (He et al., 2016; Zagoruyko & Komodakis, 2016; Vaswani et al., 2017; Sutskever et al., 2014; Goodfellow et al., 2014; Ho et al., 2020). Nevertheless, they always struggle with generalization when faced with distribution shifts in test data, which is a common challenge in real-world scenarios. For instance, natural images sourced from diverse geographic locations, timeframes, and angles inherently exhibit variations in appearance, such as differences in brightness and contrast. Similarly, medical images acquired through various devices may vary due to differences in imaging protocols.

---

\*Jinhui Xu and Ziyun Huang are co-corresonding authors.
†Part of Jinhui Xu's work was done at the University at Buffalo, SUNY.

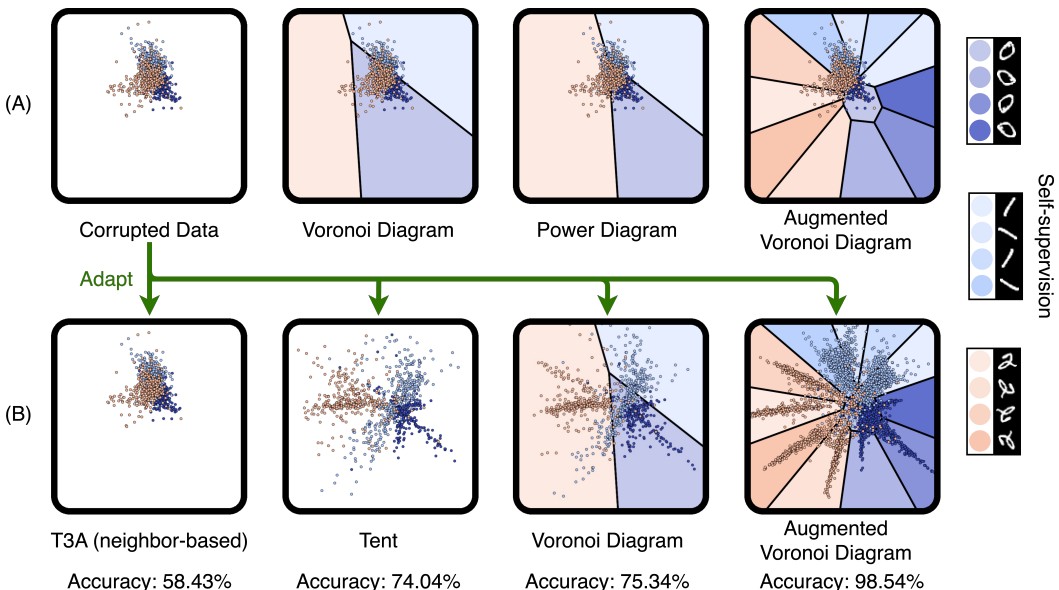

Figure 1: (A) Visualization of space partitions induced by Voronoi Diagram, Power Diagram and Augmented Voronoi Diagram (by self-supervision) on MNIST-C (Mu & Gilmer, 2019) (digit "0" ∼ "2" only, gaussian-noise-corrupted) in $\mathbb{R}^2$. (B) Visualization of adaptation performance on MNIST-C using T3A (Iwasawa & Matsuo, 2021), Tent (Wang et al., 2021) and VD and Augmented VD with joint influence. See Appendix C for details.

Test-time adaptation (TTA) (Wang et al., 2021; Sun et al., 2020; Liu et al., 2021b; Niu et al., 2023; Iwasawa & Matsuo, 2021; Zhang et al., 2021; Gong et al., 2022; Wang et al., 2022; Goyal et al., 2022; Niu et al., 2022; Zhao et al., 2023) has emerged as an online adaptation strategy to tackle the problem. While TTA shares some similarities with domain adaptation (French et al., 2017; Ganin et al., 2016), it differs in two key aspects: the source data is unavailable at test time, and only the current mini-batch of unlabeled test data is used for adaptation. Recent studies on TTA have primarily focused on two categories of methods: self-supervision, as proposed by Sun et al. (2020); Liu et al. (2021b), and entropy minimization, as proposed by Wang et al. (2021). Despite these advances, current TTA methods still face two critical limitations as follows.

**(I)** The first challenges is the reliance on insufficient or incomplete information during test-time, which restricts the ability of these methods to fully adapt to unseen data. For instance, self-supervision may inadvertently lead to overfitting on auxiliary tasks, which in turn degrades the model's performance on the primary objective, such as object recognition (Liu et al., 2021b). Additionally, more recent work (Press et al., 2024) points out that entropy minimization may fail after many iterations due to test feature embeddings drifting from the training data class means. In response to these challenges, neighbor-based methods (Liang et al., 2020; Jang et al., 2022; Liang et al., 2021; Zhang et al., 2023; Hardt & Sun, 2024) have gained attention in recent state-of-the-art approaches, as they leverage information from the training data neighborhood to mitigate overfitting and align test embedding. However, these methods often fail to adjust the model sufficiently to learn better patterns (Figure 1), resulting in suboptimal performance, and leaving the issue of robust and effective test-time adaptation unresolved. **(II)** A second critical challenge arises from negative model updates, which stem from two main factors: noisy samples and conflicting gradients. Niu et al. (2023) highlights that noisy samples can adversely affect entropy minimization, leading to suboptimal adaptation. Moreover, Gandelsman et al. (2022) demonstrates that jointly training self-supervision and entropy minimization can degrade accuracy on the ImageNet validation set due to negative transfer (Jiang et al., 2023; Javaloy & Valera, 2022). This often occurs when conflicting gradients happens from sharing a single set of network parameters for multiple task objectives, ultimately leading to diminished performance. Neighbor-based methods often handle these issues poorly due to their inherent limitations in addressing noisy samples and conflicting objectives.

In essence, the underlying geometric structure of these neighbor-based methods is *Voronoi Diagram* (VD) (Aurenhammer, 1991), a classical geometry model for space partition. This geometric framework has been applied across various domains of deep learning due to its inherent mathematical benefits (Ma et al., 2022; 2023; You et al., 2022; Balestriero et al., 2023). VD offers high interpretability, with visualizations derived from its construction algorithm in $\mathbb{R}^2$, allowing for analytical solutions to all partition boundaries (Figure 1). Additionally, recent advancements in geometric structures (Aurenhammer, 1987; Chen et al., 2013; 2017; Huang et al., 2021a) based on VD offer improved properties over its original form, creating more complex space partitions.

Building on the strengths of geometric structures, in this paper, we revisit the TTA problem from geometric view and utilize their potential to address the challenges by introducing our proposed framework, *Test-time adjustment by Voronoi Diagram guidance* (TTVD). Specifically, we focus on two key structures, *Cluster-induced Voronoi Diagram* (CIVD, (Chen et al., 2013; 2017; Huang et al., 2021a)) and the *Laguerre–Voronoi Diagram* (a.k.a *Power Diagram*, PD (Aurenhammer, 1987)). (I) CIVD, a recent breakthrough in computational geometry, extends VD from a point-to-point distance-based diagram to a cluster-to-point influence-based structure. It enables us to assign partitions (Voronoi cells) not only based on a point (e.g class prototypes), but also a cluster of points, thereby enhancing robustness during test time. (II) PD generalizes VD to create more flexible partitions by weighting each cell differently. This weighted structure enables PD to handle varying levels of influence for different points, making it particularly effective in identifying noisy samples near decision boundaries. Our contributions are summarized as follows,

- We revisit the Test-Time Adaptation problem from geometric view and formulate it using Voronoi Diagram. It is a powerful structure with two key advantages: (I) VD is highly interpretable, allowing for clear visualizations and analytical boundary solutions in $\mathbb{R}^2$, and (II) advancements in VD-based structures offer robust partitioning, which have not yet been explored in TTA. Based on these insights, we first introduce the foundation of guiding TTA by VD, paving the way to integrate more advanced geometric structures to further adaptation improvements.
- We propose to use Cluster-induced Voronoi Diagram, a recent breakthrough geometric structure to guide TTA. Specifically, extending the traditional VD to CIVD allows us to create more robust space partitions, as Voronoi cells are determined by a cluster of points rather than individual points. Furthermore, the joint influence mechanism of its cluster-to-point structure can unify multiple objectives, enables a seamless integration of self-supervision and entropy minimization, thereby improving adaptation in dynamic test environments.
- We conducted a fine-grained analysis of loss landscape utilizing the iterpretability of VD, uncovering that current sample filtering strategies may not effectively remove noisy samples. To address this, we propose to filter samples near partition boundaries by incorporating the Power Diagram. PD's flexible boundaries allow for more precise identification of noisy samples, thereby improving the efficiency of sample filtering and enhancing model robustness.

## 2 RELATED WORK

**Domain Adaptation.** Domain adaptation (DA.) (French et al., 2017; Ganin et al., 2016; Li et al., 2018) aims to alleviate the performance degradation caused by the distribution discrepancies between training and testing data. Classical approaches involve joint optimization on both source and target domains to enable domain generalization (Ganin et al., 2016; Li et al., 2018). Source-free domain adaptation (SFDA) (Liang et al., 2020; 2021; Kundu et al., 2020; Liu et al., 2021a) is a subset of DA where source data is unavailable during adaptation. This setting has been explored in various studies, including SHOT (Liang et al., 2020), USFDA (Kundu et al., 2020).SFDA methods can be roughly categorized into self-supervised training (Achituve et al., 2021; Pan et al., 2020; Chen et al., 2020), neighborhood clustering (Yang et al., 2023; 2021), and adversarial alignments (Tang & Jia, 2020; Kang et al., 2018; Shen et al., 2024).

**Test-time Adaption and its Neighbor-based Methods.** Test-Time Adaptation refers to the process of adapting a pre-trained model to distribution shifts encountered during testing, without accessing the original training data. Unlike domain adaptation, which focuses on both source and target domains during training, TTA operates solely at test time, making it more flexible for real-world applications where training data may no longer be available. Many approaches to TTA have focused on neighbor-based methods, which utilize neighborhood information for adaptation. For example,

Test-Time Template Adjuster (T3A, (Iwasawa & Matsuo, 2021)) adjusts the classifier by updating the linear layer with pseudo-prototype representations derived from the test data. Similarly, Test-Time Adaptation via Self-Training (TAST, (Jang et al., 2022)) introduces trainable adaptation modules on top of a frozen feature extractor, while AdaNPC (Zhang et al., 2023) leverages deep nearest neighbor classifiers for adaptation. In addition to these neighbor-based methods, other approaches explore TTA from different perspectives, including self-training (Sun et al., 2020; Liu et al., 2021b) and entropy minimization (Wang et al., 2021; Gong et al., 2022; Niu et al., 2023; Wang et al., 2022). It is worth noting that these methods are not always mutually exclusive; many TTA techniques combine multiple strategies to improve performance, blending ideas from neighbor-based adaptation with self-training or entropy-based optimization. Some previous algorithms (e.g. SHOT (Liang et al., 2020)) in DA can also be repurposed and adapted to be used in TTA.

**Computational Geometry for Deep Learning.** Computational geometry is a branch of computer science and mathematics that focuses on the study of algorithms for solving geometric problems. Traditionally, it has been applied in computer graphics, robotics, and geographic information systems. However, recent studies have increasingly explored its potential in machine learning, uncovering valuable insights across various domains. In the theoretical perspective, studies (Balestriero et al., 2019; Balestriero & Baraniuk, 2021) have established a link between convolutional neural networks and computational geometry, showing that deep networks recursively partition the input space into cells. Similarly, (Wang et al., 2019) provides a geometric analysis of recurrent neural networks (RNNs), demonstrating that RNNs also segment input space. More recently, (Balestriero et al., 2023) reveals that the output of a multi-head attention block in transformers is the Minkowski sum of convex hulls, enabling feature extraction for downstream tasks. In the application perspective, computational geometry has been utilized in deep learning for various tasks. DeepVoro (Ma et al., 2022) unifies few-shot learning methods using Cluster-induced Voronoi Diagrams (Huang et al., 2021b) to aggregate heterogeneous features. iVoro (Ma et al., 2023) enables exemplar-free class-incremental learning by progressively constructing Voronoi cells for new classes. SplineCam (Humayun et al., 2023) visualizes decision boundaries and input partitions using continuous piecewise linear splines. FedUD Lei et al. (2025) addresses the domain gap in federated learning by ensuring consistency in Voronoi Diagrams across clients, effectively aligning feature distributions.

## 3  METHODOLOGY

In this section, we first revisit the general setting of TTA. Then, we introduce the geometric framework based on the Voronoi Diagram and further extend it to two well-established geometric structures, the Power Diagram and the Cluster-induced Voronoi Diagram.

**Problem Setup.** Test-time adaptation refers to the process of adapting a pre-trained model to distribution shifts that occur between the training and testing phases, without accessing the original training data or labels during test time. Formally, let $\mathcal{D}_{train}$ and $\mathcal{D}_{test}$ be the training and test distributions, respectively, where $\mathcal{D}_{test}$ exhibits a shift from $\mathcal{D}_{train}$. The goal of TTA is to adapt the model $f_\theta$, with parameters $\theta$ learned from $\mathcal{D}_{train}$, using only the unlabeled test data $\mathcal{X}_{test}$ to improve performance on the shifted distribution. For a $K$-way classification problem, online test stream of data $\{x_t\} \in \mathcal{X}_{test}$ are used to update the model $\theta$ as follows at every time step $t$,

$$\text{infer:} \quad \tilde{y}_t = f_{\theta_t}(x_t), \quad \text{adapt:} \quad \theta_{t+1} = \theta_t - \lambda \nabla \mathcal{L}(\tilde{y}_t) \tag{1}$$

where $\tilde{y}_t$ represents the model's prediction for $x_t$, and $\mathcal{L}$ is the user-defined loss function. For example, Tent (Wang et al., 2021) minimizes the entropy loss $\mathcal{L} = -\sum p(\tilde{y}_t) \log p(\tilde{y}_t)$, while TTT (Sun et al., 2020) minimizes the self-supervised rotation prediction loss from the auxiliary classifier. Commonly, only the channel-wise affine parameters in normalization layers are updated during TTA, while the rest of the model remains unchanged. This approach ensures computational efficiency, making it suitable for real-time adaptation during testing. For convenience in notation and throughout the following analysis, the parameter set $\theta$ is separated into two components: the feature extractor, denoted as $\sigma$, and the classifier, denoted as $\psi$. The time step subscript $t$ is dropped unless otherwise specified.

## 3.1 VORONOI DIAGRAM: FOUNDATIONAL GEOMETRIC STRUCTURE FOR NEIGHBOR-BASED TEST-TIME ADAPTATION

Geometrically, Voronoi Diagram has long been a foundational structure for the analysis of nearest neighbor algorithms. It partitions space based on distances to a set of points as follows,

**Definition 3.1** (Voronoi Diagram). Let $d$ be the distance function associated with $\mathbb{R}^\ell$, where $\ell$ is the dimensionality of feature space. A Voronoi Diagram partitions the space into $K$ disjoint cells $\Omega = \{\omega_1, \cdots, \omega_K\}$ such that $\cup_{r=1}^{K} \omega_r = \mathbb{R}^\ell$. Each cell is obtained via $\omega_r = \{z \in \mathbb{R}^\ell : r(z) = r\}$, $r \in \{1, \cdots, K\}$, with

$$r(z) = \underset{k \in \{1, \cdots, K\}}{\arg\min} \ d(z, \mu_k), \tag{2}$$

where $\mu_k$ is the center (also referred to as Voronoi site) of $k$-th cell. VD partitions into the space $K$ disjoint cells, where the boundaries between these cells are determined by the distances that are equidistant from two or more sites. These boundaries form the edges of the Voronoi cells, and they help to define distinct regions around each site. Based on this property, VD can classify feature points by Equation 2, assigning each point to the site that minimizes the distance between them. In TTA, since the training distribution $\mathcal{D}_{test}$ deviates from $\mathcal{D}_{train}$, feature points may not fall into correct cells (Figure 1). Therefore, at every time step, the adaptation can be formulated based on alignments between feature points and Voronoi cells, with our propsed VD-based loss,

$$\textit{infer:} \quad \tilde{y}_k = \beta(-d(\sigma(x), \mu_k) + \epsilon; \tau), \quad \textit{VD loss:} \quad \mathcal{L}_{\text{VD}}(\tilde{y}_k) = -\sum_k \tilde{y}_k \log \tilde{y}_k \tag{3}$$

where $\beta(z_j; \tau) = \frac{e^{\frac{z_j}{\tau}}}{\sum_j e^{\frac{z_j}{\tau}}}$ is a softmax function with temperature scaling factor $\tau$, $\epsilon$ is the machine epsilon for improving numerical stability in code implementation and $\tilde{y}_k$ is the predicted soft label of $x$. The intuition behind this distance-based loss is to

---

**Algorithm 1:** VD-based Guidance for Test-time Adaptation

**Input:** Pretrained feature extractor $\sigma_0$, Voronoi sites $\mu$, test stream $\{x\}_t$
**Output:** Prediction stream $\{\tilde{y}_k\}_t$
**for** *each online batch* $\{x\}_t$ **do**
  *infer:* $\tilde{y}_k = \beta(-d(\sigma(x), \mu_k) + \epsilon; \tau)$ ;   // Equation 3
  *adapt:* $\sigma_{t+1} = \sigma_t - \lambda \nabla \mathcal{L}_{\text{VD}}(\tilde{y}_t)$ ;   // Equation 1
**end**

---

encourage feature points to move closer to one of the Voronoi sites. The scaling factor $\tau$ controls the regulation strength towards the sites. When a feature point is sufficiently close to a site, the VD loss is minimized. This formulation can be seamlessly integrated into TTA, as presented in Algorithm 1, forming the basis for more advanced geometric structures that will be introduced later. Commonly, the Voronoi site can be set using the class mean of the training data $\mathcal{X}_{train}$.

## 3.2 CLUSTER-INDUCED VORONOI DIAGRAM: MULTI-SITE INFLUENCES MECHANISM IMPROVES ROBUSTNESS

Cluster-induced Voronoi Diagram is a generalization of the ordinary Voronoi Diagram that extends VD from a point-to-point distance-based diagram to a cluster-to-point influence-based structure. While VD has been extensively studied for its exceptional utility in a wide range of analyses, its inherent simplicity can be limiting in certain complex scenarios. One key characteristic of VD is that the influence from each site is independent and does not interact or combine with other sites. However, in real-world applications, it is common for influences from multiple sources to be "combined" to create a joint influence. For example, in physics, a point mass $p$ may receive forces from a number of other masses, and the combined effect of these forces jointly determines the motion of $p$. CIVD improves VD by introducing such a multi-source influence as below,

**Definition 3.2** (Cluster-induced Voronoi Diagram (Chen et al., 2013; 2017; Huang et al., 2021b)). Let $\mathcal{C} = \{\mathcal{C}_1, \ldots, \mathcal{C}_K\}$ be a set of cluster and $F(z, \mathcal{C}_k)$ is a pre-defined influence function. A Cluster-induced Voronoi Diagram partitions the space into $K$ disjoint cells $\Omega = \{\omega_1, \cdots, \omega_K\}$ such that $\cup_{r=1}^{K} \omega_r = \mathbb{R}^\ell$. Each cell is obtained via $\omega_r = \{z \in \mathbb{R}^\ell : r(z) = r\}$, $r \in \{1, \cdots, K\}$, with $r(z) = \underset{k \in \{1, \ldots, K\}}{\arg\max} \ F(z, \mathcal{C}_k)$, where the influence between $z$ and $\mathcal{C}_k = \{\mu_k^{(\alpha)}\}$ are commonly defined

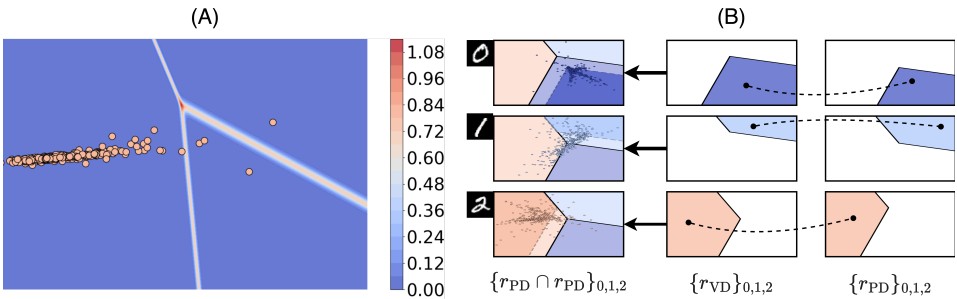

Figure 2: Noisy sample filtering by diagram subtraction. (a) Entropy landscape of MNIST. Loss value quickly shrinks once a sample leave the boundaries. (b) Multi-site provides more reliable samples. The solid and dash line are boundaries given by PD and VD, respectively. Reliable samples can be identified by subtracting Voronoi cells, marked in deeper colors.

as

$$F(z, \mathcal{C}_k) = -\operatorname{sign}(\gamma) \sum_\alpha (d(\mu_k^{(\alpha)}, z))^\gamma. \tag{4}$$

Here, $\alpha$ denotes the item index of the cluster $\mathcal{C}_k$ and $\gamma$ is a hyperparameter that controls the scale of the influence. Similar to VD, CIVD partitions the space into $K$ disjoint cells, while the boundaries are determined by a cluster of points $\mathcal{C}_k$, given the influence function $F$ (Equation 4). Inspired by this, CIVD shows great promise for robust adaptation through its multi-source influence mechanism, offering greater effectiveness in scenarios where a single-point influence is insufficient. It is particularly well suited for TTA, where only small batches of data are available at each time step. The multi-source framework allows the model to dynamically adapt to the limited information provided, improving its ability to generalize and maintain performance in challenging, real-time settings where traditional methods may struggle to capture the full complexity of the data distribution. Specifically, $\mathcal{C}_k$ can be established via self-supervision, benefiting from data augmentation for improved robustness. We utilize rotation augmentation, where images are rotated at 4 different angles $\operatorname{Rot}_\alpha \in \{0, 90, 180, 270\}$ to generate $\mathcal{C}_k$, and each rotation corresponds to a Voronoi site $\mu_k^{(\alpha)}$. This process is performed using self-supervised label augmentation (Lee et al., 2020). Similar to Equation 3, the soft label given by CIVD can be calculated from the influence function, incorporating the expanded sites $\mu_k^{(\alpha)}$, enhancing robustness against individual predictions.

Additionally for TTA, CIVD expands Voronoi site $\mu_k$ to a cluster of site $\mathcal{C}_k$, integrating the approach of self-supervision and entropy minimization. The joint label $\tilde{y}_k^{(\alpha)}$ avoids the negative transfer since the objective is now unified.

### 3.3 POWER DIAGRAM: IDENTIFYING NOISY SAMPLES BY FLEXIBLE BOUNDARIES

Laguerre–Voronoi Diagram (a.k.a Power Diagram) is another generalization of the Voronoi Diagram that extends the concept by moving from equally-weighted sites to variably-weighted sites. In traditional VD, each site is treated equally, which may not be suitable for all scenarios. PD improves VD by introducing the power distance between a point and a site as follows,

**Definition 3.3** (Power Diagram (Aurenhammer, 1987)). Let $d$ be the distance function associated with space $\mathbb{R}^\ell$, a Power Diagram partitions the space into $K$ disjoint cells $\Omega = \{\omega_1, \cdots, \omega_K\}$ such that $\cup_{r=1}^K \omega_r = \mathbb{R}^\ell$. Each cell is associated with a weight $v_k$ and is obtained via $\omega_r = \{z \in \mathbb{R}^\ell : r(z) = r\}, r \in \{1, \cdots, K\}$, with

$$r(z) = \underset{k \in \{1, \cdots, K\}}{\arg\min} \, d(z, \mu_k)^2 - v_k^2. \tag{5}$$

**Lemma 3.1** (Ma et al. (2022; 2023)). A logistic regression model parameterized by $W^{K \times \ell}$ and $b^K$ partitions the feature space $\mathbb{R}^\ell$ into a $K$-cell Power Diagram with $\mu_k = \frac{1}{2}W^{k \times \ell}$ and $v_k^2 = b^k + \frac{1}{4}\left\|W^{k \times \ell}\right\|_2^2$.

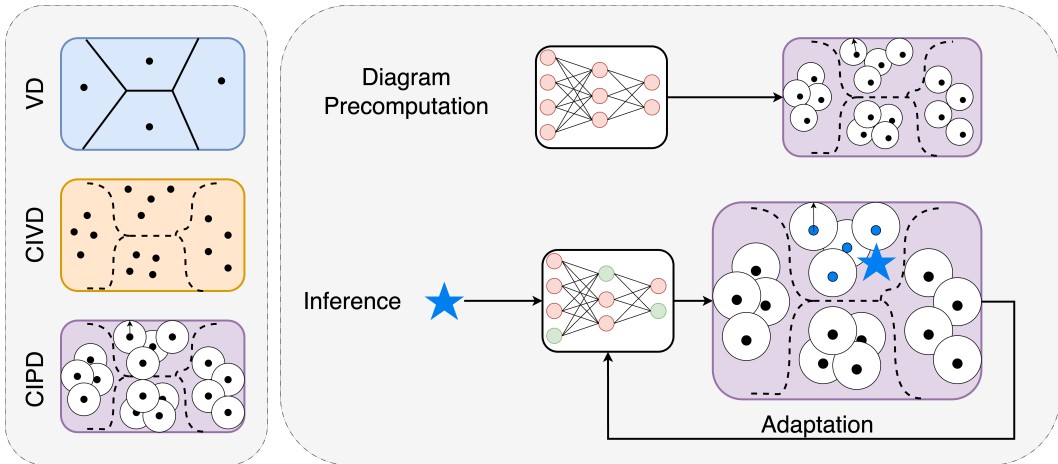

Figure 3: (Left) Illustrations on differences between VD, CIVD and CIPD. (Right) Illustrations on Test-time adaptations by Voronoi Diagram(s) guidance.

An illustration of the Power Diagram is given in Figure 1. By adding weights to the sites, the boundaries of the cells can be shifted in orthogonal directions, allowing for more flexible partitioning. Noted that CIVD and PD are parallel structures, meaning they can be seamlessly integrated. CIVD can be retrofitted to CIPD as follows for further robustness improvements,

**Definition 3.4** (Cluster-induced Power Diagram). Let $\mathcal{C} = \{\mathcal{C}_1, \ldots, \mathcal{C}_K\}$ be a set of cluster and $F(z, \mathcal{C}_k)$ is a pre-defined influence function. a Cluster-induced Power Diagram partitions the space into $K$ disjoint cells $\Omega = \{\omega_1, \cdots, \omega_K\}$ such that $\cup_{r=1}^{K} \omega_r = \mathbb{R}^\ell$. Each cell is obtained via $\omega_r = \{z \in \mathbb{R}^\ell : r(z) = r\}$, $r \in \{1, \cdots, K\}$, with $r(z) = \arg\max_{k \in \{1,\ldots,K\}} F(z, \mathcal{C}_k)$, where the influence between $z$ and $\mathcal{C}_k = \{\mu_k^{(\alpha)}\}$ are defined as

$$F(z, \mathcal{C}_k) = -\operatorname{sign}(\gamma) \sum_{\alpha} \{d(\mu_k^{(\alpha)}, z)^2 - v_k^2\}^\gamma. \tag{6}$$

As mentioned earlier, noisy samples negatively impact entropy minimization, resulting in suboptimal adaptation. Existing methods propose addressing this issue by filtering out these samples based on their entropy values, drawing from empirical observations of the relationship between adaptation accuracy and gradient norms. This approach is plausible since models tend to be more confident in predicting low-entropy samples, and the gradients produced by these samples are considered more reliable. However, the underlying relationship between entropy values and sample selection remains unclear. To further explore this, we adopt a geometric perspective using the interpretability of the VD. From the visualization of the entropy loss landscape in Figure 2a, it can be observed that noisy samples are only identifiable if they are near the boundaries, leaving many noisy samples undetected. Inspired by the boundary-shifting capability of the PD, we propose incorporating PD to improve noisy sample filtering. By subtracting the PD from the VD, we can extract a larger region from the resulting differences, which may also capture areas contributing to unstable gradients. Noisy samples in these regions are excluded during adaptation, thereby enhancing the robustness of the model.

Overall, our proposed TTVD is constructed progressively, transitioning from standard VD to CIVD and CIPD, as summarized in Figure 3. At testing-time, we infer and adapt the model accordingly by CIPD (Algorithm 3 in Appendix H) using Equation 6.

## 4 EXPERIMENTS

In this section, we present a comprehensive evaluation of our method, benchmarking it against other approaches using the peer-reviewed, open-source toolkit TTAB (Zhao et al., 2023), a standardized codebase designed to ensure fair comparisons across methods.

## 4.1 EXPERIMENT SETUP

**Dataset.** CIFAR-10-C, CIFAR-100-C, and ImageNet-C (Hendrycks & Dietterich, 2019) are benchmark datasets designed to assess model robustness in the presence of various corruptions and shift. CIFAR-10-C and CIFAR-100-C are corrupted versions of the original CIFAR-10 and CIFAR-100 datasets, where each image has been subjected to 15 different types of common corruptions such as noise, blur, and weather distortions, with five levels of severity. ImageNet-C applies similar corruptions to the large-scale ImageNet dataset, providing a higher-resolution challenge for models. ImageNet-R (ImageNet-Renditions, (Hendrycks et al., 2021)) consists of non-photorealistic renditions of ImageNet classes, such as paintings, cartoons, and sculptures, testing a model's ability to generalize beyond traditional photographic imagery. These datasets allow us to comprehensively assess the robustness of our method under a range of real-world distortions and domain shifts.

**Compared Methods.** We include the four groups of state-of-the-art methods for the experiments listed below, and their extended introduction are given in Appendix F.

- Neighbor-based methods: (I) T3A (Iwasawa & Matsuo, 2021), (II) TAST (Jang et al., 2022).
- Repurposed domain adaptation methods: (I) BN_Adapt (Schneider et al., 2020), (II) SHOT (Liang et al., 2020).
- Self-training methods: TTT (Sun et al., 2020).
- Entropy-based methods: (I) TENT (Wang et al., 2021), (II) NOTE (Gong et al., 2022), (III) Conjugate_PL (Goyal et al., 2022), (IV) SAR (Niu et al., 2023).

**Implementation Details.** We adhere to the standard settings given in TTAB for fairness comparison. Specifically, generic hyperparameters are grid-searched for the best combination, following guidelines in TTA. Method-specific hyperparameters for each TTA algorithm are selected according to their original experimental setups. Results are reported using the optimal configuration for each method. For TTVD, we trained ResNet-26 for CIFAR-10-C and CIFAR-100-C, and ResNet-50 for ImageNet-C and ImageNet-R, following the official recipe from the torchvision library, using label augmentation (Lee et al., 2020). We use the full training set of CIFAR-10, CIFAR-100 to compute the class means for Voronoi sites and 10% of ImageNet for similar calculation.

**Evaluation Metrics.** Two metrics are used to report the performance: classification error and expected calibration error (ECE) on online test samples. ECE measures the trustworthiness of the model's confidence in its predictions, which is crucial in real-world applications.

## 4.2 EXPERIMENT RESULTS

**Overall Performance Comparison.** TTVD demonstrates the best overall performance across multiple datasets. Even under rigid grid-search tuning, our method consistently achieves the lowest classification error and ECE, reducing classification errors by 0.8%, 0.7%, 1.6%, 0.7% on the four datasets, respectively, and ECE by 3.4%, 1.8%, 4.1% and 4.3%, demonstrating its trustworthiness.

**Effect of Components in TTVD.** We ablate our methods by gradually downgrading CIPD to the very basic VD. From Table 2, the performance of VD already surpasses that of other neighbor-based methods. When generalizing VD to CIVD, we observe a significant improvement of 5.7% overall for all corruption types. across all corruption types. To investigate the reason behind this, we conducted a sample-level analysis in Appendix A.1, which demonstrates that the multi-influence structure of CIVD enhances its robustness. Finally, CIPD, with its flexible boundaries and noise filtering mechanisms, further improves upon CIVD by an additional 2.2%, showcasing its superior adaptability.

---

[1]The subscripted values represent comparisons made under the oracle model selection setting from TTAB. These values may not reflect real-world performance, as they assume access to ground truth test labels to select optimal models during test time—a condition rarely available in practical scenarios. Additionally, it has been shown from TTAB that, in some cases, this approach can lead to overfitting to online batches. While these results may indicate optimal performance in controlled environments, they do not accurately represent how the model would perform in real-world, label-free settings.

[2]The experimental settings of TTAB are followed to omit the values for TTT on the ImageNet dataset. This omission aligns with the TTAB guidelines for fair comparison across methods.

Table 1: Comparison of State-of-the-art Methods Regarding **Error** (♣) and **Expected Calibration Error** (♦). The Top Optimal Results are Highlighted in **Bold**.[1]Metrics are Reported Using Level-5 Corruption for CIFAR-C and ImageNet-C, Averaged over 15 Corruption Types. Detailed Results for Each Corruption Type are Provided in Appendix B.

| | CIFAR10-C(%)↓ | | CIFAR100-C(%)↓ | | ImageNet-C(%)↓ | | ImageNet-R(%)↓ | |
| --- | --- | --- | --- | --- | --- | --- | --- | --- |
| | ♣ | ♦ | ♣ | ♦ | ♣ | ♦ | ♣ | ♦ |
| T3A(Iwasawa & Matsuo, 2021) | 40.3 | 19.5 | 67.6 | 21.1 | 83.1 | 26.3 | 79.4 | 20.5 |
| TAST(Jang et al., 2022) | 39.6 | 40.5 | 69.8 | 29.2 | 74.8 | 25.1 | 78.8 | 21.1 |
| BN_Adapt(Schneider et al., 2020) | 27.5 | 18.1 | 56.6 | 18.5 | 72.3 | 32.8 | 68.9 | 30.9 |
| SHOT(Liang et al., 2020) | 21.9$_{(21.0)}$ | 16.4 | 49.8$_{(46.8)}$ | 18.5 | 63.4$_{(62.4)}$ | 36.4 | 68.6 | 31.2 |
| TTT(Sun et al., 2020)[2] | 21.3$_{(20.0)}$ | 15.2 | 53.4$_{(51.9)}$ | 20.2 | ——— | | | |
| TENT(Wang et al., 2021) | 24.0$_{(21.7)}$ | 16.9 | 53.5$_{(49.9)}$ | 18.3 | 62.7$_{(61.9)}$ | 38.7 | 68.3 | 31.4 |
| NOTE(Gong et al., 2022) | 28.6$_{(24.0)}$ | 21.5 | 58.5$_{(54.5)}$ | 23.5 | 65.7$_{(69.8)}$ | 34.1 | 68.2 | 31.7 |
| Conjugate PL(Goyal et al., 2022) | 24.0$_{(22.9)}$ | 16.9 | 53.5$_{(51.0)}$ | 18.3 | 63.1$_{(62.2)}$ | 38.4 | 68.7 | 31.2 |
| SAR(Niu et al., 2023) | 24.2$_{(21.9)}$ | 16.9 | 53.7$_{(49.7)}$ | 18.1 | 61.4$_{(59.1)}$ | 38.4 | 68.5 | 31.3 |
| **TTVD** (Ours) | **20.5**$_{(20.0)}$ | **11.8** | **49.1**$_{(49.0)}$ | **17.0** | **59.8**$_{(58.2)}$ | **21.0** | **67.5** | **16.8** |

Table 2: Ablation Study Using Different Geometric Structures on CIFAR-10-C Across Various Corruption Types Regarding Error (%)↓.

| | Noise | | | Blur | | | Weather | | | Digital distortion | | | | | |
| --- | --- | --- | --- | --- | --- | --- | --- | --- | --- | --- | --- | --- | --- | --- | --- |
| | gau | sho | imp | def | gla | mot | zoo | sno | fro | bri | con | ela | fog | pix | jpg | Avg. |
| VD | 37.5 | 34.5 | 43.8 | 19.5 | 42.9 | 25.6 | 21.2 | 26.5 | 25.6 | 15.0 | 20.0 | 30.1 | 23.3 | 27.3 | 33.5 | 28.4 |
| CIVD | 30.0 | 27.0 | 35.9 | 14.8 | 36.2 | 19.7 | 16.0 | 21.5 | 20.0 | 11.6 | 15.9 | 24.6 | 17.8 | 21.1 | 27.9 | 22.7$_{(↓5.7)}$ |
| CIPD | **27.4** | **24.6** | **32.8** | **13.2** | **36.0** | **18.1** | **14.2** | **19.9** | **17.5** | **10.1** | **13.2** | **22.6** | **15.3** | **18.2** | **24.6** | **20.5**$_{(↓2.2)}$ |

**Adaptation Curves.** As discussed in earlier sections regarding the phenomenon of model over-fitting in TTA, it is imperative to thoroughly investigate the adaptation dynamics as the adaptation process unfolds over time. As presented in Figure 4, Tent and SAR do not show signs of overfitting. This may be due to the rigid experimental settings and thorough grid search process we employed, ensuring optimal hyperparameter selection. However, it can be observed that TTVD consistently outperforms across the four noise throughout the entire sequence of online batches. The model maintains a significant downward trend over the various time steps, suggesting that it continues to learn and adapt effectively, with the potential for further improvements if provided with more data. This highlights TTVD's robustness and resilience against overfitting.

In contrast, both TENT and SAR exhibit more modest improvements in adaptation, and their performance often stagnates or converges at lower accuracy levels compared to TTVD. Specifically, SAR shows a notable limitation in its ability to adapt, particularly in the presence of impulse noise, where it quickly reaches a plateau and ceases to improve. Furthermore, in the case of defocus blur, SAR struggles to learn useful patterns

Table 3: Comparison to Neighbor-based Methods Regarding Error (%)↓ on Four types of Blur Corruption in ImageNet-C.

| | Defoc | Glass | Motion | Zoom |
| --- | --- | --- | --- | --- |
| T3A(Iwasawa & Matsuo, 2021) | 92.2 | 90.3 | 90.7 | 85.2 |
| TAST(Jang et al., 2022) | 83.7 | 92.0 | 92.3 | 76.7 |
| AdaNPC(Zhang et al., 2023) | 83.1 | 83.0 | 72.3 | 60.6 |
| **TTVD** | **79.5** | **77.7** | **68.6** | **53.2** |

in the early stages of adaptation, resulting in poor performance on the initial batches. TENT, while slightly better than SAR in some cases, also demonstrates limitations in adapting to these perturbations. The early stagnation of both Tent and SAR may indicate potential overfitting to specific noise conditions or a failure to effectively generalize across different noise types as TTVD does.

**Comparison to Neighbor-based Methods.** We follow the report of an additional nearest neighbor method, AdaNPC (Zhang et al., 2023), to benchmark our method in four types of blur corruption in ImageNet-C (defocus blur, glass blur, motion blur and zoom blur). In Table 3, TTVD consistently outperforms the previous methods, demonstrating superior robustness to blur distortions.

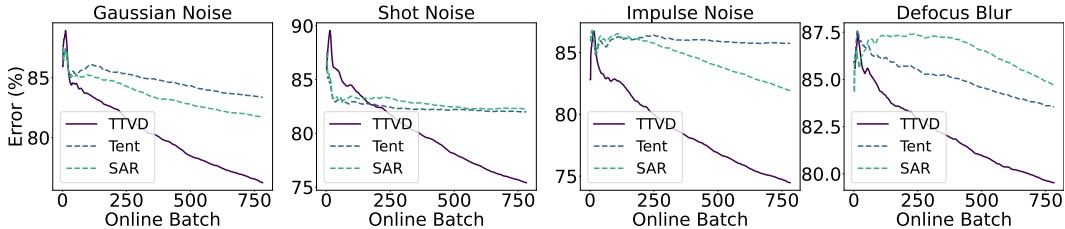

Figure 4: Comparison on the Adaptation Curves on different noise perturbations in ImageNet-C. Error (%)↓ is calculated over all retrospective test samples. The first four types of perturbations in the dataset are presented above.

**How Accurate Should the Class Means Be?** TTVD requires offline calculation of Voronoi sites, which must be performed during the pre-training phase. In our experiments, this calculation took less than 10 minutes on 10% of the ImageNet training set using an NVIDIA-RTX A6000. However, in the new era of large-scale datasets, this process may become more resource-intensive. Interestingly, TTVD demonstrates high robustness to the precision of these Voronoi sites, as shown in Table 4.

Table 4: Robustness to Class Mean Precision Using Different Proportions of ImageNet Data.

|  | 10% | 5% | 1% |
|---|---|---|---|
| **TTVD** | 59.8 | 59.8 | 59.9 |

**Effect of Batch Size and Label Shift.** Test-time adaptation often receives small batches every time, and label shift, i.e., Non iid test stream may happen in online adaptation. We tested TTVD with various smaller batch sizes and different level of label shifted data in Appendix B, demonstrating its high ability to adapt under challenging scenarios.

## 5 CONCLUSION

In this paper, we revisit the Test-Time Adaptation problem from a geometric perspective, formulating it using the Voronoi Diagram—a classical and powerful structure in computational geometry known for its elegant mathematical properties. Building on the foundation of guiding TTA with traditional Voronoi Diagram, we extend the approach to more advanced geometric structures, namely the Cluster-induced Voronoi Diagram and the Power Diagram. These structures offer enhanced flexibility and robustness, making them particularly well-suited for TTA. Our experiments demonstrate the effectiveness of our proposed method, TTVD, across a variety of datasets and scenarios, highlighting its capacity to adapt to diverse challenges in real-world settings.

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

## A APPENDIX

### A.1 SAMPLE ANALYSIS

In Section 4.2, experimental results indicate that CIVD contributes the most to the improvement. To understand the reason behind this, we investigate how misclassified samples are corrected after CIVD is employed. We arbitrarily inspect three examples from the "bike", "bus", and "clock" class in Figure 5, Figure 6 and Figure 7, respectively. The distances between the feature points and all Voronoi sites are shown.

- The "bike" example originally is misclassified as "lobster" in an individual VD. However, the 90-degree rotated image is correctly classified. When the CIVD applies the influence function to aggregate the information of all four rotations, the model eventually gets the correct prediction.
- In the "bus" example, all four rotated images are misclassified as various classes, such as "bowl", "table" or "house". However, the distances to the ground-true label are all relatively small. CIVD aggregates these distances and makes the correct prediction. The "clock" example shows a similar phenomenon.

In conclusion, this sample analysis reveals that the expanded Voronoi sites and rotated image set contribute to the improvement in CIVD.

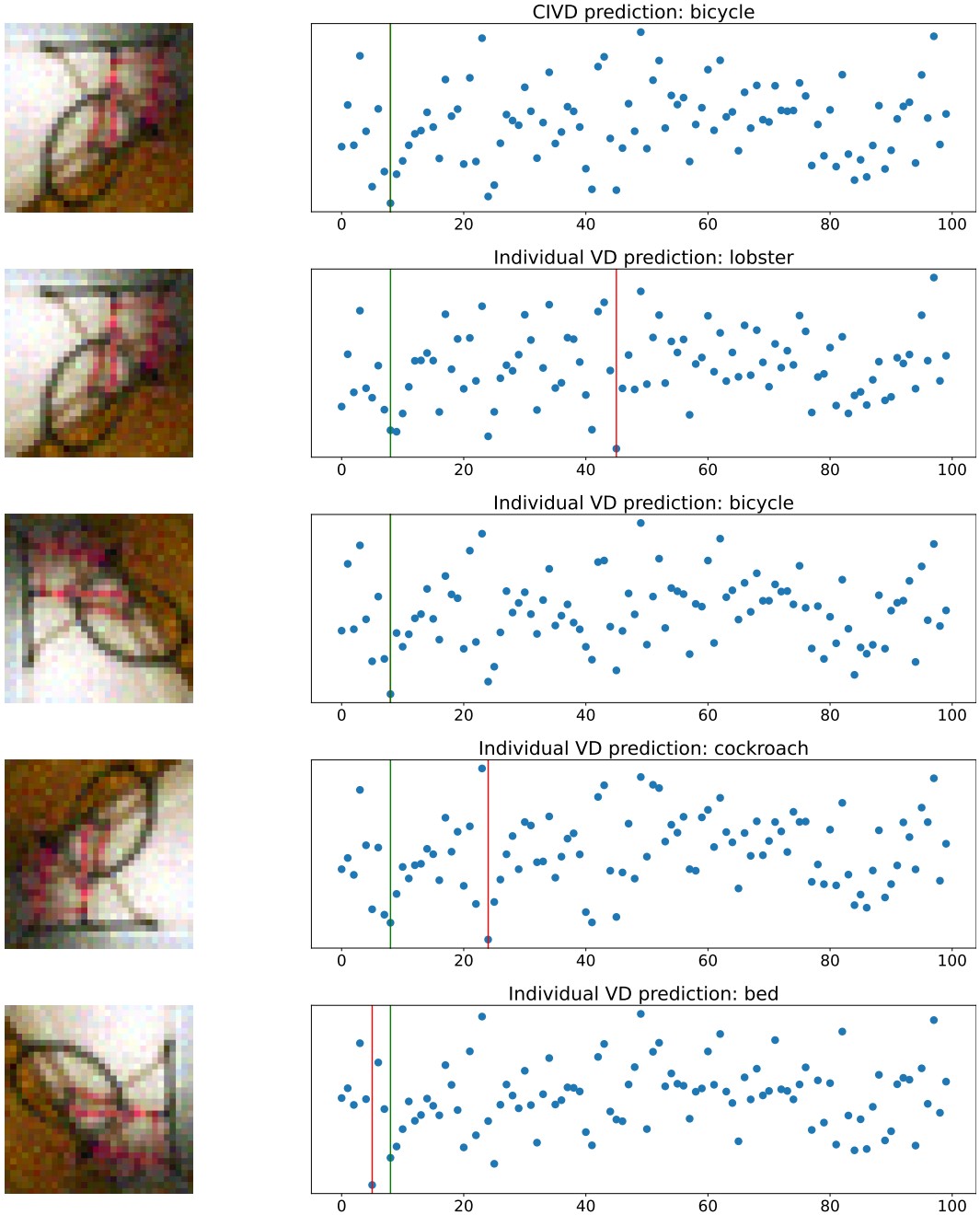

Figure 5: A misclassified "bike" sample corrected by CIVD. The x-axis denotes the index of classes and y-axis denotes the distance to their corresponding Voronoi sites. The green lines indicate the ground-true label and the red lines indicate the predicted label.

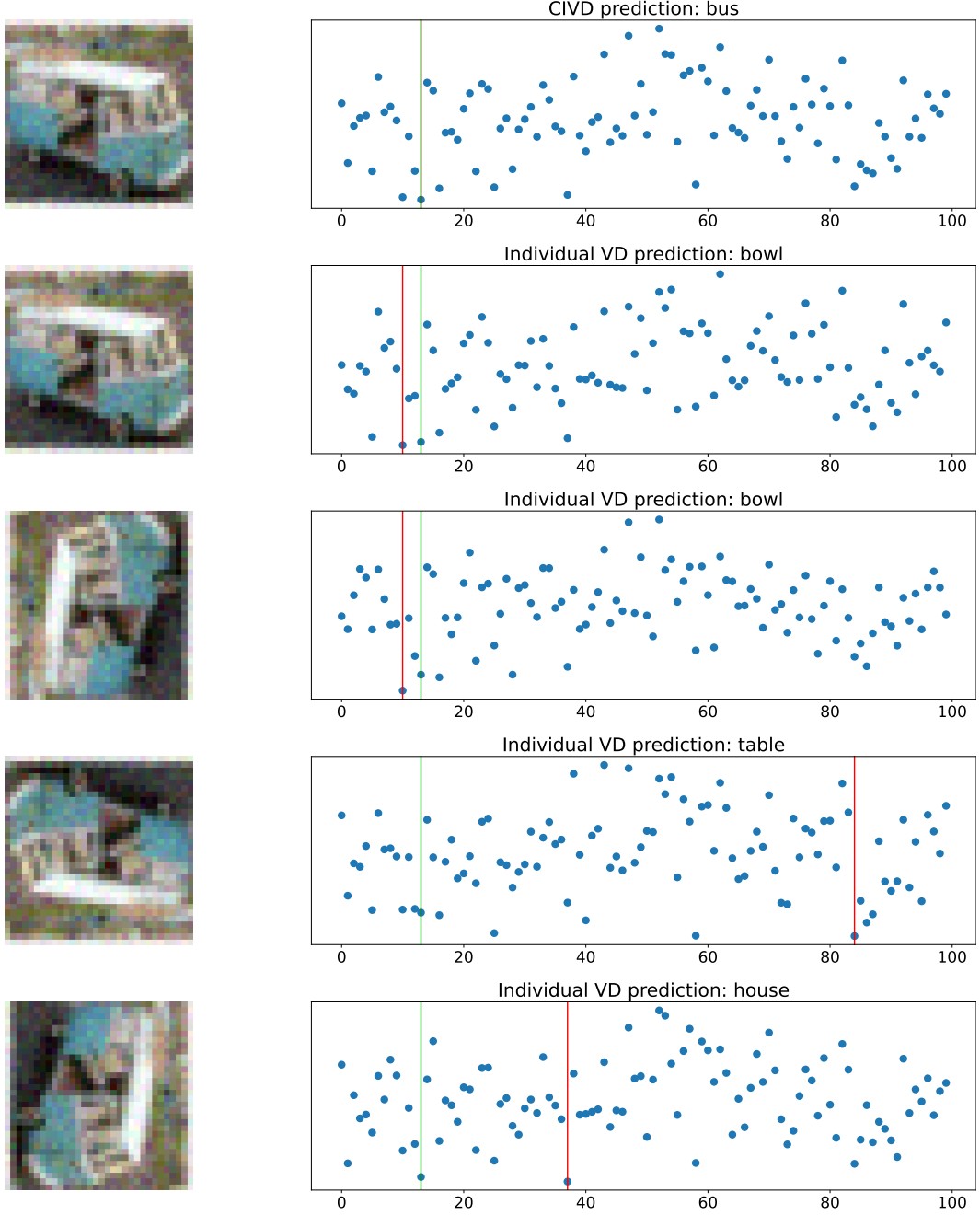

Figure 6: A misclassified "bus" sample corrected by CIVD. The x-axis denotes the index of classes and y-axis denotes the distance to their corresponding Voronoi sites. The green lines indicate the ground-true label and the red lines indicate the predicted label.

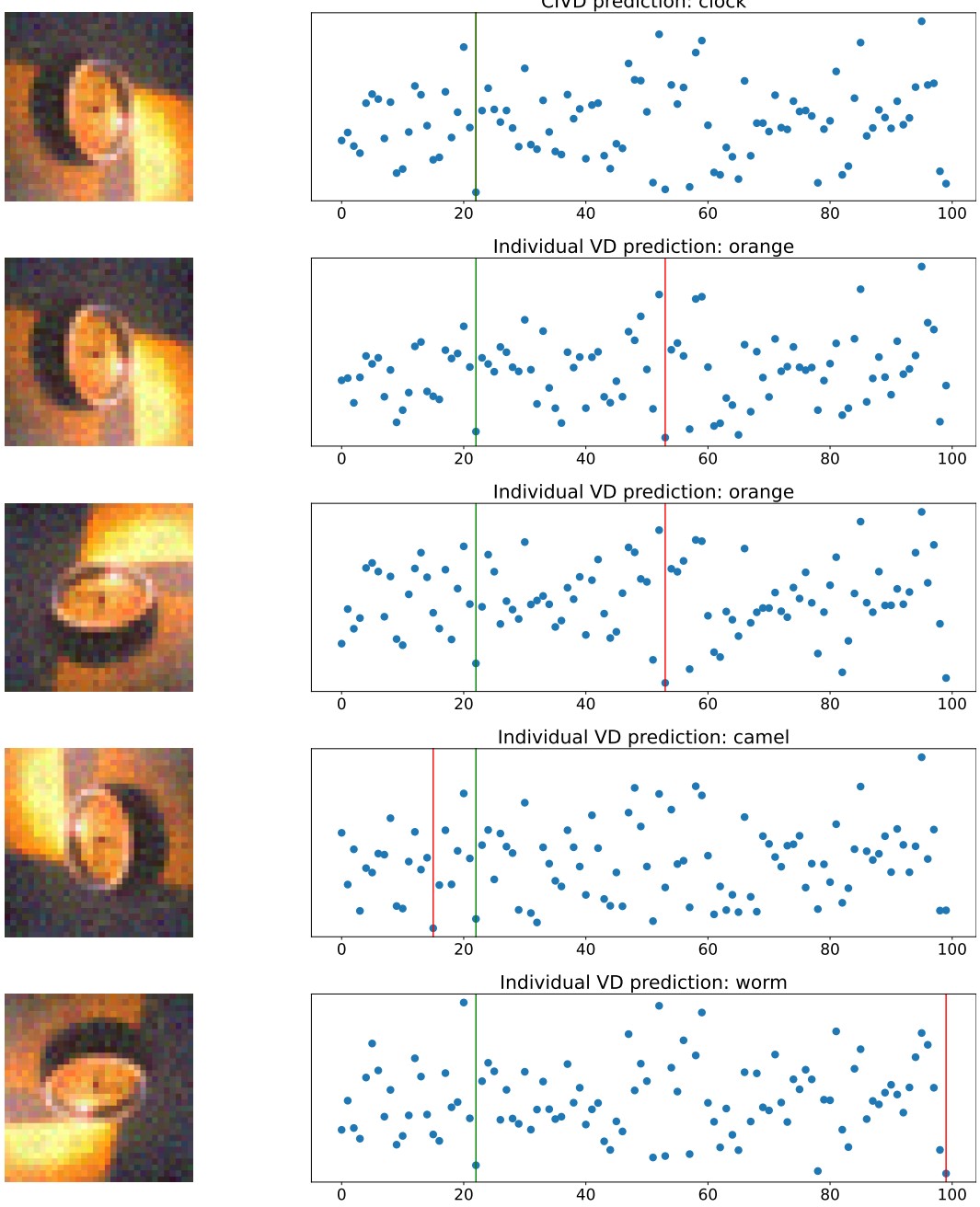

Figure 7: A misclassified "clock" sample corrected by CIVD. The x-axis denotes the index of classes and y-axis denotes the distance to their corresponding Voronoi sites. The green lines indicate the ground-true label and the red lines indicate the predicted label.

# B ADDITIONAL TABLES

Table 5: Comparison Regarding Error (%)↓ on CIFAR10-C Level-5.

| | Noise | | | Blur | | | | Weather | | | | Digital distortion | | | | |
| | gau | sho | imp | def | gla | mot | zoo | sno | fro | fog | bri | con | ela | pix | jpg | Avg. |
|---|---|---|---|---|---|---|---|---|---|---|---|---|---|---|---|---|
| T3A | 65.1 | 59.5 | 65.3 | 37.0 | 47.5 | 34.5 | 33.8 | 24.7 | 36.4 | 34.5 | 10.1 | 49.8 | 25.1 | 51.6 | 29.7 | 40.3 |
| TAST | 63.0 | 58.3 | 64.5 | 36.7 | 46.9 | 33.6 | 33.0 | 24.7 | 35.6 | 34.1 | 10.3 | 49.8 | 25.0 | 49.2 | 29.6 | 39.6 |
| BN_Adapt | 39.2 | 37.0 | 46.0 | 17.3 | 41.3 | 19.9 | 17.6 | 25.2 | 25.4 | 20.5 | 14.0 | 17.8 | 29.1 | 26.5 | 35.5 | 27.5 |
| SHOT | 29.3 | 27.0 | 34.7 | 14.2 | 33.6 | 16.8 | 15.0 | 19.2 | 21.6 | 18.1 | 11.5 | 16.1 | 25.4 | 20.1 | 26.5 | 21.9 |
| TTT | 25.6 | 23.0 | 29.8 | 13.2 | 34.6 | 20.0 | 15.6 | 19.8 | 17.7 | 14.0 | 9.2 | 26.1 | 24.0 | 16.0 | 23.2 | 21.3 |
| TENT | 32.5 | 29.7 | 39.2 | 15.6 | 36.9 | 18.1 | 16.1 | 21.4 | 23.0 | 19.3 | 12.6 | 16.9 | 26.5 | 22.7 | 29.9 | 24.0 |
| NOTE | 47.3 | 40.1 | 43.9 | 23.3 | 38.1 | 22.6 | 21.3 | 19.8 | 23.4 | 21.3 | 9.2 | 30.7 | 23.8 | 35.9 | 27.9 | 28.6 |
| Conjugate_PL | 32.5 | 29.8 | 39.2 | 15.6 | 36.9 | 18.0 | 16.1 | 21.4 | 23.0 | 19.3 | 12.6 | 16.9 | 26.5 | 22.7 | 29.9 | 24.0 |
| SAR | 33.3 | 30.0 | 39.7 | 15.7 | 36.8 | 18.0 | 16.1 | 21.9 | 22.8 | 19.4 | 12.7 | 17.2 | 26.4 | 22.8 | 30.5 | 24.2 |
| TTVD | 27.4 | 24.6 | 32.8 | 13.2 | 36.0 | 18.1 | 14.2 | 19.9 | 17.5 | 15.3 | 10.1 | 13.2 | 22.6 | 18.2 | 24.6 | **20.5** |

Table 6: Comparison Regarding Error (%)↓ on CIFAR100-C Level-5.

| | Noise | | | Blur | | | | Weather | | | | Digital distortion | | | | |
| | gau | sho | imp | def | gla | mot | zoo | sno | fro | fog | bri | con | ela | pix | jpg | Avg. |
|---|---|---|---|---|---|---|---|---|---|---|---|---|---|---|---|---|
| T3A | 89.3 | 88.4 | 90.4 | 64.8 | 60.7 | 59.8 | 57.2 | 57.0 | 61.1 | 65.6 | 43.4 | 82.3 | 50.1 | 82.7 | 60.6 | 67.6 |
| TAST | 89.2 | 88.2 | 90.7 | 66.3 | 63.5 | 63.1 | 60.0 | 62.1 | 64.7 | 67.9 | 47.8 | 82.4 | 55.0 | 81.6 | 64.1 | 69.8 |
| BN_Adapt | 70.3 | 70.1 | 69.1 | 46.5 | 61.1 | 48.8 | 45.9 | 58.9 | 56.6 | 55.1 | 45.1 | 51.0 | 53.2 | 54.2 | 62.6 | 56.6 |
| SHOT | 58.4 | 57.6 | 59.1 | 41.2 | 55.2 | 44.2 | 41.2 | 51.9 | 50.6 | 48.7 | 40.2 | 49.0 | 48.7 | 46.2 | 55.4 | 49.8 |
| TTT | 64.0 | 63.2 | 65.5 | 43.8 | 57.4 | 49.6 | 43.3 | 54.7 | 50.5 | 49.6 | 38.8 | 70.0 | 49.5 | 45.6 | 56.4 | 53.4 |
| TENT | 65.1 | 64.6 | 65.0 | 44.1 | 58.0 | 46.9 | 43.4 | 55.9 | 54.4 | 52.4 | 42.5 | 49.4 | 51.6 | 50.3 | 59.5 | 53.5 |
| NOTE | 76.1 | 74.3 | 74.6 | 53.8 | 57.5 | 50.8 | 47.6 | 52.5 | 51.8 | 56.1 | 38.8 | 67.1 | 48.6 | 70.5 | 57.6 | 58.5 |
| Conjugate_PL | 65.1 | 64.6 | 65.0 | 44.1 | 58.1 | 46.8 | 43.4 | 55.9 | 54.4 | 52.4 | 42.5 | 49.4 | 51.7 | 50.4 | 59.5 | 53.5 |
| SAR | 65.3 | 64.9 | 65.2 | 44.2 | 58.3 | 47.2 | 47.6 | 56.5 | 54.6 | 52.4 | 42.6 | 48.7 | 51.7 | 50.5 | 59.6 | 53.7 |
| TTVD | 58.2 | 57.4 | 63.2 | 38.8 | 59.9 | 45.7 | 40.2 | 50.7 | 49.3 | 45.7 | 36.6 | 42.1 | 50.6 | 44.1 | 54.4 | **49.1** |

Table 7: Comparison Regarding Error (%)↓ on ImageNet-C Level-5.

| | Noise | | | Blur | | | | Weather | | | | Digital distortion | | | | |
| | gau | sho | imp | def | gla | mot | zoo | sno | fro | fog | bri | con | ela | pix | jpg | Avg. |
|---|---|---|---|---|---|---|---|---|---|---|---|---|---|---|---|---|
| T3A | 89.0 | 87.7 | 89.8 | 91.7 | 90.9 | 90.2 | 84.7 | 78.7 | 97.8 | 70.2 | 43.2 | 85.5 | 94.9 | 88.6 | 74.5 | 83.1 |
| TAST | 81.0 | 79.6 | 82.0 | 83.7 | 92.0 | 82.3 | 76.7 | 70.2 | 69.4 | 62.4 | 34.3 | 77.6 | 86.6 | 89.6 | 55.3 | 74.8 |
| BN_Adapt | 91.1 | 88.0 | 91.6 | 92.0 | 90.9 | 79.1 | 65.9 | 64.7 | 63.4 | 47.0 | 36.7 | 81.5 | 62.5 | 65.9 | 64.8 | 73.1 |
| SHOT | 75.2 | 72.0 | 76.3 | 86.7 | 85.0 | 75.2 | 61.9 | 52.7 | 53.4 | 38.7 | 29.9 | 96.2 | 51.4 | 47.6 | 48.4 | 63.4 |
| TENT | 83.4 | 80.2 | 83.9 | 83.5 | 81.7 | 68.3 | 56.3 | 55.9 | 55.2 | 39.1 | 29.9 | 69.9 | 52.8 | 50.3 | 50.6 | 62.7 |
| NOTE | 80.4 | 77.3 | 80.7 | 86.4 | 85.4 | 73.0 | 59.8 | 58.5 | 56.0 | 42.1 | 29.8 | 77.6 | 57.0 | 66.7 | 55.0 | 65.7 |
| Conjugate_PL | 84.0 | 80.7 | 84.6 | 83.9 | 83.4 | 68.8 | 56.6 | 56.5 | 55.7 | 39.2 | 29.9 | 71.0 | 52.9 | 49.5 | 49.8 | 63.1 |
| SAR | 81.7 | 82.3 | 81.9 | 84.7 | 82.1 | 65.1 | 54.4 | 54.1 | 54.0 | 38.5 | 29.7 | 66.8 | 50.2 | 47.8 | 48.3 | 61.4 |
| TTVD | 76.2 | 75.4 | 74.4 | 79.5 | 77.7 | 68.6 | 53.2 | 55.9 | 58.7 | 41.2 | 30.4 | 65.0 | 47.3 | 42.1 | 50.8 | **59.8** |

Table 8: Comparison Regarding Expected Calibration Error (%)↓ on CIFAR10-C Level-5.

| | Noise | | | Blur | | | | Weather | | | | Digital distortion | | | | |
|---|---|---|---|---|---|---|---|---|---|---|---|---|---|---|---|---|
| | gau | sho | imp | def | gla | mot | zoo | sno | fro | fog | bri | con | ela | pix | jpg | Avg. |
| T3A | 13.3 | 14.9 | 13.3 | 22.9 | 19.2 | 20.3 | 23.4 | 21.7 | 19.7 | 18.9 | 21.4 | 21.3 | 23.5 | 16.1 | 21.7 | 19.5 |
| TAST | 18.2 | 22.9 | 17.6 | 43.3 | 33.7 | 46.2 | 46.6 | 54.4 | 43.8 | 45.7 | 68.1 | 31.4 | 54.7 | 30.7 | 50.4 | 40.5 |
| BN_Adapt | 24.8 | 23.4 | 28.8 | 12.4 | 26.4 | 13.5 | 12.8 | 17.2 | 16.5 | 14.1 | 10.3 | 11.2 | 19.1 | 17.4 | 22.8 | 18.1 |
| SHOT | 21.5 | 19.5 | 25.4 | 11.1 | 24.4 | 12.7 | 11.2 | 14.3 | 15.9 | 13.8 | 9.0 | 13.3 | 18.8 | 15.0 | 19.4 | 16.4 |
| TTT | 18.3 | 16.5 | 20.4 | 10.5 | 23.8 | 14.4 | 12.0 | 14.4 | 13.1 | 10.7 | 7.7 | 20.2 | 17.3 | 12.0 | 16.5 | 15.2 |
| TENT | 22.3 | 20.4 | 26.8 | 11.7 | 25.1 | 13.0 | 11.8 | 15.2 | 15.9 | 14.1 | 9.6 | 11.9 | 18.7 | 16.1 | 20.6 | 16.9 |
| NOTE | 34.9 | 31.1 | 30.8 | 18.1 | 28.2 | 17.6 | 16.9 | 14.6 | 16.9 | 17.3 | 7.7 | 20.4 | 17.1 | 31.0 | 20.5 | 21.5 |
| Conjugate_PL | 22.2 | 20.4 | 26.9 | 11.7 | 25.1 | 13.0 | 11.9 | 15.2 | 16.0 | 14.2 | 9.5 | 11.9 | 18.6 | 16.0 | 20.5 | 16.9 |
| SAR | 22.4 | 20.5 | 26.8 | 11.8 | 24.8 | 12.9 | 11.9 | 15.4 | 16.0 | 14.0 | 9.3 | 11.8 | 18.6 | 16.1 | 20.9 | 16.9 |
| TTVD | 13.8 | 12.9 | 15.4 | 9.9 | 15.9 | 11.4 | 9.8 | 11.6 | 11.1 | 10.6 | 8.3 | 9.3 | 12.4 | 11.4 | 13.4 | **11.8** |

Table 9: Comparison Regarding Expected Calibration Error (%)↓ on CIFAR100-C Level-5.

| | Noise | | | Blur | | | | Weather | | | | Digital distortion | | | | |
|---|---|---|---|---|---|---|---|---|---|---|---|---|---|---|---|---|
| | gau | sho | imp | def | gla | mot | zoo | sno | fro | fog | bri | con | ela | pix | jpg | Avg. |
| T3A | 7.9 | 8.6 | 6.8 | 21.4 | 25.8 | 25.8 | 27.7 | 28.6 | 25.4 | 22.6 | 35.0 | 10.4 | 33.0 | 10.7 | 26.6 | 21.1 |
| TAST | 9.8 | 10.8 | 8.3 | 32.6 | 35.4 | 35.8 | 39.0 | 36.8 | 34.3 | 31.0 | 51.1 | 16.5 | 43.9 | 17.3 | 34.9 | 29.2 |
| BN_Adapt | 21.2 | 21.0 | 21.3 | 16.2 | 19.7 | 17.0 | 15.6 | 20.1 | 18.4 | 17.5 | 16.1 | 17.4 | 17.7 | 18.1 | 20.4 | 18.5 |
| SHOT | 19.7 | 19.8 | 20.0 | 16.2 | 20.6 | 17.1 | 15.5 | 18.9 | 18.9 | 17.3 | 16.1 | 21.5 | 18.6 | 17.4 | 19.9 | 18.5 |
| TTT | 22.4 | 22.7 | 23.0 | 17.1 | 21.3 | 18.2 | 16.6 | 21.2 | 18.9 | 18.2 | 15.6 | 32.2 | 18.2 | 17.6 | 20.0 | 20.2 |
| TENT | 20.0 | 20.5 | 20.5 | 16.3 | 19.5 | 16.8 | 15.1 | 19.5 | 18.4 | 17.5 | 16.0 | 18.7 | 17.9 | 17.3 | 19.8 | 18.3 |
| NOTE | 32.1 | 31.3 | 29.8 | 20.4 | 21.6 | 19.0 | 18.8 | 20.8 | 20.9 | 21.5 | 16.1 | 28.9 | 18.4 | 32.7 | 20.6 | 23.5 |
| Conjugate_PL | 20.0 | 20.5 | 20.5 | 16.3 | 19.5 | 16.8 | 15.1 | 19.5 | 18.4 | 17.5 | 16.0 | 18.7 | 17.9 | 17.3 | 19.9 | 18.3 |
| SAR | 20.2 | 20.3 | 20.6 | 16.2 | 19.9 | 16.6 | 15.4 | 19.7 | 18.1 | 17.4 | 16.0 | 16.8 | 17.9 | 17.1 | 19.5 | 18.1 |
| TTVD | 12.2 | 12.8 | 11.0 | 22.5 | 12.6 | 18.4 | 22.2 | 15.7 | 16.0 | 19.4 | 23.8 | 18.1 | 16.5 | 19.2 | 15.2 | **17.0** |

Table 10: Comparison Regarding Expected Calibration Error (%)↓ on ImageNet-C Level-5.

| | Noise | | | Blur | | | | Weather | | | | Digital distortion | | | | |
|---|---|---|---|---|---|---|---|---|---|---|---|---|---|---|---|---|
| | gau | sho | imp | def | gla | mot | zoo | sno | fro | fog | bri | con | ela | pix | jpg | Avg. |
| T3A | 20.4 | 21.7 | 19.6 | 17.7 | 9.0 | 19.2 | 24.7 | 30.7 | 31.6 | 39.2 | 66.2 | 23.9 | 14.5 | 11.3 | 45.4 | 26.3 |
| TAST | 18.9 | 20.3 | 17.9 | 16.3 | 7.9 | 17.6 | 23.2 | 29.7 | 30.5 | 37.5 | 65.6 | 22.3 | 13.3 | 10.3 | 44.6 | 25.1 |
| BN_Adapt | 14.1 | 17.2 | 13.6 | 13.2 | 14.3 | 26.0 | 39.3 | 40.4 | 41.7 | 58.2 | 68.4 | 23.7 | 42.6 | 39.2 | 40.4 | 32.8 |
| SHOT | 24.6 | 27.8 | 23.6 | 13.1 | 14.8 | 24.7 | 37.9 | 47.1 | 46.4 | 61.1 | 69.9 | 3.7 | 48.4 | 52.2 | 51.4 | 36.4 |
| TENT | 17.9 | 21.8 | 17.9 | 18.1 | 20.5 | 33.2 | 44.8 | 45.4 | 46.2 | 61.5 | 70.1 | 31.0 | 48.3 | 51.9 | 51.3 | 38.7 |
| NOTE | 19.4 | 22.6 | 19.2 | 13.5 | 14.5 | 26.9 | 40.0 | 41.3 | 43.9 | 57.7 | 69.9 | 22.2 | 42.8 | 33.1 | 44.8 | 34.1 |
| Conjugate_PL | 17.1 | 20.4 | 16.8 | 17.1 | 20.6 | 34.0 | 45.0 | 45.1 | 45.8 | 61.6 | 70.3 | 28.6 | 49.8 | 53.0 | 51.6 | 38.4 |
| SAR | 18.1 | 17.6 | 18.0 | 15.2 | 17.8 | 34.7 | 45.4 | 45.8 | 45.8 | 61.3 | 70.1 | 33.1 | 49.6 | 52.0 | 51.5 | 38.4 |
| TTVD | 10.4 | 11.0 | 11.1 | 8.9 | 9.7 | 14.7 | 24.9 | 23.0 | 22.4 | 33.7 | 41.8 | 16.7 | 28.5 | 31.5 | 26.1 | **21.0** |

Table 11: Comparison Regarding Error (%)↓ on ImageNet-C Level-5 with Various Smaller Batch Size.

| | Noise | | | Blur | | | | Weather | | | | Digital distortion | | | | |
| --- | gau | sho | imp | def | gla | mot | zoo | sno | fro | fog | bri | con | ela | pix | jpg | Avg. |
| Batch-Size-32 | | | | | | | | | | | | | | | | |
| T3A | 79.5 | 78.2 | 80.3 | 82.2 | 90.9 | 80.7 | 75.2 | 69.2 | 68.3 | 60.7 | 33.8 | 76.0 | 85.3 | 88.7 | 54.5 | 73.6 |
| BN_Adapt | 86.2 | 83.4 | 86.9 | 87.2 | 86.4 | 75.1 | 62.1 | 60.8 | 59.2 | 43.1 | 32.9 | 77.8 | 59.0 | 62.2 | 61.0 | 68.2 |
| SHOT | 75.1 | 72.0 | 76.0 | 91.5 | 87.5 | 77.6 | 63.3 | 54.3 | 55.4 | 40.1 | 31.2 | 98.8 | 53.2 | 50.1 | 50.7 | 65.1 |
| TENT | 81.4 | 79.0 | 79.5 | 82.1 | 81.3 | 66.9 | 56.2 | 54.5 | 54.1 | 39.1 | 30.8 | 70.0 | 52.1 | 48.6 | 48.9 | 61.6 |
| NOTE | 82.8 | 79.8 | 83.5 | 86.0 | 84.9 | 72.6 | 59.2 | 58.3 | 56.6 | 41.2 | 30.0 | 76.0 | 56.1 | 62.6 | 56.4 | 65.7 |
| Conjugate_PL | 82.0 | 78.7 | 81.1 | 81.5 | 81.2 | 65.0 | 55.0 | 53.6 | 54.0 | 38.6 | 30.7 | 72.9 | 50.1 | 46.5 | 47.8 | 61.2 |
| SAR | 89.4 | 79.5 | 78.0 | 85.6 | 79.5 | 67.2 | 55.0 | 53.1 | 53.6 | 38.9 | 30.8 | 67.5 | 50.8 | 48.3 | 51.4 | 61.9 |
| TTVD | 80.7 | 76.1 | 73.8 | 79.5 | 78.1 | 67.3 | 52.2 | 54.6 | 59.0 | 40.6 | 30.6 | 63.1 | 47.7 | 41.2 | 49.3 | **59.6** |
| Batch-Size-16 | | | | | | | | | | | | | | | | |
| T3A | 79.5 | 78.2 | 80.3 | 82.2 | 90.9 | 80.8 | 75.2 | 69.2 | 68.3 | 60.7 | 33.7 | 76.0 | 85.4 | 88.6 | 54.5 | 73.6 |
| BN_Adapt | 87.7 | 85.0 | 88.0 | 88.5 | 88.0 | 77.7 | 66.0 | 63.1 | 61.7 | 46.1 | 35.9 | 80.0 | 62.5 | 65.3 | 64.4 | 70.7 |
| SHOT | 77.7 | 74.7 | 78.2 | 95.6 | 91.1 | 85.9 | 69.3 | 57.4 | 58.5 | 43.3 | 34.5 | 99.4 | 57.5 | 53.4 | 54.6 | 68.7 |
| TENT | 85.1 | 80.0 | 79.4 | 83.6 | 84.5 | 68.7 | 60.6 | 54.9 | 56.9 | 41.4 | 33.7 | 76.3 | 55.0 | 51.3 | 54.0 | 64.4 |
| NOTE | 84.0 | 80.8 | 84.5 | 85.8 | 84.8 | 72.5 | 59.0 | 58.3 | 56.7 | 40.6 | 30.0 | 75.3 | 55.7 | 60.4 | 57.0 | 65.7 |
| Conjugate_PL | 81.2 | 79.5 | 78.5 | 82.7 | 79.9 | 66.4 | 59.1 | 53.8 | 54.9 | 41.8 | 33.2 | 69.1 | 52.8 | 48.3 | 52.8 | 62.3 |
| SAR | 81.8 | 88.1 | 83.6 | 93.0 | 86.6 | 69.7 | 58.4 | 53.7 | 54.4 | 40.3 | 33.2 | 77.4 | 53.3 | 48.9 | 52.5 | 65.0 |
| TTVD | 74.1 | 72.4 | 72.1 | 78.6 | 77.3 | 69.0 | 54.4 | 56.4 | 58.2 | 42.5 | 32.3 | 68.1 | 48.4 | 43.7 | 51.4 | **59.9** |
| Batch-Size-8 | | | | | | | | | | | | | | | | |
| T3A | 79.5 | 78.2 | 80.3 | 82.2 | 90.9 | 80.8 | 75.2 | 69.2 | 68.3 | 60.7 | 33.7 | 76.0 | 85.4 | 88.7 | 54.5 | 73.6 |
| BN_Adapt | 89.8 | 87.7 | 89.9 | 90.8 | 90.5 | 82.1 | 72.5 | 68.2 | 66.4 | 52.6 | 42.7 | 83.4 | 69.1 | 71.8 | 70.0 | 75.2 |
| SHOT | 83.8 | 81.5 | 84.2 | 98.2 | 96.2 | 90.8 | 77.7 | 64.1 | 63.6 | 52.8 | 42.2 | 99.6 | 68.5 | 63.4 | 62.5 | 75.3 |
| TENT | 97.8 | 96.6 | 86.7 | 95.8 | 91.1 | 86.6 | 75.1 | 64.8 | 69.7 | 48.0 | 41.6 | 94.4 | 69.6 | 61.6 | 68.9 | 76.6 |
| NOTE | 84.6 | 81.4 | 85.2 | 85.7 | 84.6 | 72.4 | 58.9 | 58.3 | 56.9 | 40.3 | 30.2 | 74.7 | 55.5 | 59.3 | 57.1 | 65.7 |
| Conjugate_PL | 87.7 | 80.0 | 79.2 | 86.9 | 84.1 | 72.7 | 65.8 | 58.9 | 59.1 | 47.1 | 39.5 | 81.1 | 59.3 | 55.4 | 58.2 | 67.7 |
| SAR | 92.5 | 90.5 | 89.1 | 92.4 | 90.0 | 80.2 | 67.9 | 60.0 | 60.0 | 46.1 | 38.8 | 77.3 | 60.9 | 56.7 | 57.4 | 70.6 |
| TTVD | 78.2 | 75.9 | 76.3 | 84.0 | 83.2 | 80.0 | 58.3 | 59.5 | 62.1 | 45.6 | 35.6 | 75.8 | 51.8 | 46.8 | 54.5 | **64.5** |

Table 12: Comparison Regarding Error on ImageNet-C Level-5 with Non-i.i.d test stream, Generated by Dirichlet Distribution with Parameter $\alpha$. Lower Value of $\alpha$ Indicates Worse Label Shift.

| | Noise | | | Blur | | | | Weather | | | | Digital distortion | | | | |
| --- | gau | sho | imp | def | gla | mot | zoo | sno | fro | fog | bri | con | ela | pix | jpg | Avg. |
| $\alpha = 1$ | | | | | | | | | | | | | | | | |
| T3A | 79.6 | 78.3 | 80.3 | 82.1 | 90.8 | 80.6 | 75.2 | 69.1 | 68.3 | 60.8 | 33.7 | 75.9 | 85.3 | 88.6 | 54.6 | 73.5 |
| BN_Adapt | 85.7 | 82.5 | 86.4 | 86.5 | 85.7 | 73.8 | 60.8 | 59.6 | 58.1 | 41.8 | 31.5 | 76.4 | 57.3 | 60.6 | 59.6 | 67.1 |
| SHOT | 75.0 | 71.9 | 76.5 | 86.3 | 87.0 | 74.3 | 60.3 | 53.1 | 53.8 | 38.9 | 30.3 | 97.9 | 51.2 | 47.3 | 49.0 | 63.5 |
| TENT | 81.7 | 78.1 | 81.8 | 82.1 | 81.4 | 66.7 | 55.1 | 54.3 | 54.3 | 38.6 | 29.5 | 68.1 | 51.8 | 47.9 | 51.3 | 61.5 |
| NOTE | 80.5 | 77.4 | 80.9 | 86.6 | 85.4 | 72.6 | 60.0 | 58.4 | 56.1 | 42.2 | 29.6 | 78.2 | 56.8 | 66.6 | 54.7 | 65.7 |
| Conjugate_PL | 83.0 | 79.0 | 83.2 | 83.4 | 81.1 | 66.7 | 54.6 | 54.8 | 54.1 | 38.3 | 29.6 | 71.0 | 51.0 | 47.2 | 51.6 | 61.9 |
| SAR | 86.0 | 76.6 | 80.1 | 88.5 | 83.6 | 66.2 | 55.0 | 54.6 | 54.0 | 38.4 | 29.6 | 68.4 | 50.8 | 49.7 | 51.2 | 62.2 |
| TTVD | 77.8 | 75.0 | 74.3 | 79.1 | 77.1 | 68.7 | 53.1 | 55.9 | 57.8 | 41.2 | 30.5 | 65.7 | 47.6 | 42.6 | 50.8 | **59.8** |
| $\alpha = 0.1$ | | | | | | | | | | | | | | | | |
| T3A | 79.7 | 78.2 | 80.3 | 82.1 | 90.9 | 80.8 | 75.1 | 69.2 | 68.4 | 60.9 | 33.8 | 76.0 | 85.2 | 88.7 | 54.5 | 73.6 |
| BN_Adapt | 85.8 | 82.6 | 86.4 | 86.6 | 85.7 | 74.1 | 61.0 | 59.7 | 58.5 | 42.1 | 32.0 | 76.3 | 57.6 | 60.7 | 59.7 | 67.3 |
| SHOT | 76.0 | 72.4 | 77.2 | 88.0 | 85.4 | 76.7 | 62.4 | 53.7 | 54.5 | 39.6 | 30.6 | 98.1 | 52.8 | 47.9 | 49.8 | 64.3 |
| TENT | 82.3 | 78.4 | 82.3 | 82.5 | 81.6 | 67.2 | 55.7 | 55.2 | 54.2 | 38.9 | 30.4 | 70.0 | 52.4 | 48.8 | 48.9 | 61.9 |
| NOTE | 80.6 | 77.5 | 80.8 | 86.5 | 85.5 | 72.8 | 59.9 | 58.4 | 56.2 | 42.4 | 29.9 | 78.2 | 56.8 | 66.7 | 54.9 | 65.8 |
| Conjugate_PL | 82.6 | 79.4 | 83.2 | 82.8 | 82.4 | 67.0 | 55.4 | 55.1 | 54.7 | 38.5 | 30.1 | 70.9 | 50.8 | 47.5 | 48.8 | 62.0 |
| SAR | 85.9 | 78.6 | 81.5 | 86.5 | 83.0 | 66.6 | 55.5 | 54.7 | 54.5 | 39.1 | 30.2 | 68.8 | 52.6 | 48.3 | 49.2 | 62.3 |
| TTVD | 77.2 | 75.1 | 74.2 | 79.4 | 77.1 | 68.7 | 53.1 | 56.5 | 58.8 | 41.8 | 30.8 | 66.8 | 47.6 | 42.6 | 51.1 | **60.1** |
| $\alpha = 0.01$ | | | | | | | | | | | | | | | | |
| T3A | 79.5 | 78.2 | 80.2 | 82.0 | 90.8 | 80.6 | 75.1 | 69.1 | 68.3 | 61.0 | 34.0 | 75.9 | 85.1 | 88.5 | 54.6 | 73.5 |
| BN_Adapt | 88.1 | 85.5 | 88.5 | 89.0 | 88.6 | 79.3 | 68.6 | 66.6 | 65.4 | 51.8 | 42.9 | 80.4 | 65.5 | 69.0 | 67.7 | 73.1 |
| SHOT | 82.7 | 79.9 | 83.2 | 92.9 | 90.5 | 85.9 | 74.4 | 66.8 | 66.4 | 54.0 | 46.6 | 98.5 | 65.9 | 63.9 | 65.3 | 74.5 |
| TENT | 85.1 | 83.3 | 85.8 | 86.3 | 86.1 | 75.1 | 65.9 | 63.3 | 62.9 | 50.1 | 42.6 | 76.4 | 62.4 | 60.2 | 60.1 | 69.7 |
| NOTE | 80.6 | 77.6 | 80.9 | 86.8 | 85.5 | 72.8 | 59.8 | 58.5 | 56.4 | 42.8 | 29.7 | 78.1 | 56.8 | 66.7 | 55.2 | **65.9** |
| Conjugate_PL | 86.2 | 83.6 | 86.3 | 86.5 | 86.4 | 76.0 | 65.1 | 63.5 | 63.0 | 49.1 | 42.1 | 76.0 | 61.8 | 59.4 | 62.1 | 69.8 |
| SAR | 89.6 | 83.8 | 85.2 | 91.2 | 85.8 | 75.6 | 65.3 | 63.0 | 62.2 | 49.3 | 41.5 | 76.1 | 61.7 | 59.5 | 59.2 | 69.9 |
| TTVD | 80.7 | 80.6 | 80.2 | 83.3 | 82.7 | 75.8 | 62.4 | 64.3 | 65.3 | 50.6 | 40.6 | 75.0 | 57.1 | 52.3 | 60.2 | **67.4** |

## C    Demonstrative illustration of MNIST-C Dataset in $\mathbb{R}^2$

Figure 1 aims to illustrate how our method partitions the space for the MNIST-C (Mu & Gilmer, 2019) dataset. We use the clean MNIST dataset to train a ResNet26 backbone, followed by a linear layer with an output dimension of 2 for ease of visualizing realistic Voronoi Diagrams in $\mathbb{R}^2$. In the augmented Voronoi Diagram, self-supervision is employed to expand Voronoi sites, with feature means calculated as the locations of sites. We follow the same training recipe and hyperparameter settings as those for CIFAR-10-C. The positions of vertices in the boundaries are calculated using pyvoro (Sobolev, 2014), and cells are plotted using generativepy (McBride, 2014). The figure demonstrates that feature points are more distinctly separated in the augmented Voronoi Diagram, highlighting its superior adaptation ability. By leveraging self-supervision, it refines decision boundaries, leading to improved feature alignment and classification accuracy.

## D    Hyperparameter settings in the Experiment

We follow the TTAB codebase to grid search the learning rate from $\{0.005, 0.001, 0.0005\}$ for CIFAR dataset and $\{0.001, 0.0005, 0.0001\}$ for ImageNet dataset. We set $\gamma = -0.8$ to scale and reduce the influence of distant Voronoi sites. We use $\tau = 1$ as the standard temperature for the softmax function. For model pretraining, we follow the recipe of $\mathrm{ResNet50\text{-}Weights.IMAGENET1K\text{-}V1}$ from the $\mathrm{torchvision}$ library to train the feature extractor. The batch size is set to 64, aligning to previous studies for fair comparison. It can be observed from Table 13 that TTVD is robust to the choice of learning rates.

| Learning Rate | CIFAR-10-C (%) | CIFAR-100-C (%) | ImageNet-C (%) |
|---|---|---|---|
| 0.005 | 27.4 | 57.8 | – |
| 0.001 | 27.1 | 58.2 | 76.0 |
| 0.0005 | 26.9 | 58.1 | 76.2 |
| 0.0001 | – | – | 76.2 |

Table 13: Learning Rate vs. Accuracy for CIFAR-10, CIFAR-100, and ImageNet Datasets.

## E    Extended introduction to Voronoi Diagram

Voronoi diagrams are a fundamental tool in computational geometry that partition a given space into regions. The origins of Voronoi diagrams can be traced back to 1644, when philosopher René Descartes first considered similar ideas. However, they are named after Russian mathematician Georgy Voronoi, who formally defined and studied them in 1908. Voronoi's work (Voronoi, 1908a;b) extended earlier studies on quadratic forms and lattice structures, laying the mathematical groundwork for partitioning spaces into convex regions, now termed Voronoi cells. In a Voronoi diagram, space is divided into regions such that each region contains all points closer to a given site, or a point, than to any other site.

Over the years, Voronoi diagrams have been used to solve problems in various domains due to their ability to model spatial relationships and proximity. In computer science, they are employed in tasks such as nearest neighbor search, mesh generation, and image processing. In physics, Voronoi diagrams help in modeling the behavior of particle systems and simulating crystallization processes. In biology, they are used to understand the structure of cells and tissues, where natural divisions often resemble Voronoi partitions. In urban planning, Voronoi diagrams assist in the allocation of resources, such as determining optimal locations for services like hospitals or fire stations, where regions of influence need to be defined based on proximity.

Their versatility comes from the diagram's intrinsic ability to partition space in an efficient and meaningful way, especially when dealing with problems that involve spatial clustering or resource distribution. More recently, in machine learning and artificial intelligence, Voronoi diagrams have been applied to various fields (Ma et al., 2022; 2023; Humayun et al., 2023; Balestriero et al., 2023; You et al., 2022). This geometric approach forms the foundation of our proposed method, TTVD,

which leverages Voronoi diagrams to guide Test-Time Adaptation, ultimately leading to enhanced model performance in dynamically changing environments.

## F    EXTENDED INTRODUCTION TO COMPARED METHODS

**T3A** is a method designed to improve domain generalization by adjusting models during the test phase without requiring backpropagation or changes to the feature extractor. T3A creates pseudo-prototypes from online, unlabeled test data and adjusts the classifier by measuring the distance between test samples and these prototypes. By focusing only on the classifier's linear layer, T3A is lightweight and efficient, enhancing model performance on unseen domains while avoiding the risks of complex optimization processes.

**TAST** introduces trainable adaptation modules on top of a frozen feature extractor and generates pseudo-labels for test data using nearest neighbor information. This method improves upon existing TTA techniques by ensuring more robust adaptation in scenarios where test-time domain shifts occur.

**BN_Adapt** explores how deep learning models can become more robust to common image corruptions like blur and noise. The authors highlight that in many real-world applications, models can adapt to recurring corruptions using unsupervised methods. By modifying batch normalization statistics during inference, the paper demonstrates that adapting to corrupted data significantly boosts model robustness, surpassing baseline performance across several benchmarks. This simple yet effective strategy improves the performance of models on corrupted image datasets.

**SHOT** addresses unsupervised domain adaptation (UDA) without requiring access to source data, a key limitation in existing UDA methods. SHOT leverages a pre-trained source model and transfers its knowledge to the target domain by freezing the classifier module (source hypothesis) and adapting the feature extraction module for the target domain using self-supervised learning and information maximization.

**TTT** involves updating the model at test time using a self-supervised learning task on each individual test sample before making a prediction. By using tasks like image rotation prediction as the auxiliary self-supervised task, TTT allows the model to adapt better to the test distribution.

**TENT** Entropy minimization in the TENT method works by reducing the uncertainty of a model's predictions during test-time. This is done by minimizing the entropy, or uncertainty, of the predicted probability distribution. Specifically, TENT updates the model's parameters—focusing on the affine transformations in normalization layers—based on the gradient of the entropy with respect to these parameters. By iteratively adjusting the model in response to test data, TENT improves the model's confidence in its predictions without needing labeled data, resulting in better adaptation to new or corrupted data at test time.

**NOTE** aims to address challenges in adapting models to non-i.i.d. test data streams, common in real-world applications like autonomous driving. NOTE includes two components: Instance-Aware Batch Normalization (IABN), which adjusts for out-of-distribution instances, and Prediction-Balanced Reservoir Sampling (PBRS), which simulates i.i.d. samples from temporally correlated data.

**Congugate PL** leverages the convex conjugate of the training loss to create a new TTA loss function. The authors demonstrate that meta-learning the optimal TTA loss consistently recovers a function similar to the softmax-entropy for classifiers trained with cross-entropy. For models trained with other losses, such as squared loss or PolyLoss, the optimal TTA loss differs. By interpreting this through the lens of convex conjugates, the paper presents a general framework for designing TTA losses.

**SAR** investigates the challenges of TTA when faced with real-world distribution shifts, such as mixed shifts, small batch sizes, and imbalanced label distributions. The authors find that traditional batch normalization can destabilize TTA, proposing instead the use of group and layer normalization for better stability. To further enhance stability, they introduce a sharpness-aware and reliable entropy minimization method that removes noisy samples and encourages robust model updates under challenging test scenarios.

**AdaNPC** constructs a memory bank containing features and labels from the source domain, and during inference, it retrieves the nearest neighbors from this memory to predict labels for incoming test samples. This memory is dynamically updated with test features and predictions, making the method effective for handling distribution shifts.

## G    EXPERIMENTS COMPUTE RESOURCES

All experiments are conducted using GPU NVIDIA RTX A6000.

## H    ALGORITHMS

---

**Algorithm 2:** CIVD Guidance for Test-time Adaptation

---

**Input:** Pretrained feature extractor $\sigma_0$, a set of Voronoi sites $\mathcal{C}$, test stream $\{x\}_t$
**Output:** Prediction stream $\{\tilde{y}_k\}_t$
**for** *each online batch $\{x\}_t$* **do**
 $\quad$ *infer:* $\quad \tilde{y}_k = \beta(-F(z, \mathcal{C}_k) + \epsilon; \tau)$ ;                    `// Equation 4`
 $\quad$ *adapt:* $\quad \sigma_{t+1} = \sigma_t - \lambda \nabla \mathcal{L}_{\text{VD}}(\tilde{y}_t)$ ;                    `// Equation 1`
**end**

---

---

**Algorithm 3:** CIPD Guidance for Test-time Adaptation

---

**Input:** Pretrained feature extractor $\sigma_0$, a set of Voronoi sites $\mathcal{C}$, weights of Voronoi sites $v$, test
 $\quad\quad$ stream $\{x\}_t$
**Output:** Prediction stream $\{\tilde{y}_k\}_t$
**for** *each online batch $\{x\}_t$* **do**
 $\quad$ *infer:* $\quad \tilde{y}_k = \beta(-F(z, \mathcal{C}_k) + \epsilon; \tau)$ ;                    `// Equation 6`
 $\quad$ *adapt:* $\quad \sigma_{t+1} = \sigma_t - \lambda \nabla \mathcal{L}_{\text{VD}}(\tilde{y}_t)$ ;                    `// Equation 1`
**end**

---

