# OpenReview forum: "TTVD: Towards a Geometric Framework for Test-Time Adaptation Based on Voronoi Diagram"
_ICLR.cc/2025/Conference — ICLR 2025 Poster_

### Official Review · Reviewer_wGB6 · 2024-10-30

**Soundness:** 3
**Presentation:** 3
**Contribution:** 2
**Rating:** 6
**Confidence:** 3

**Summary:**

This paper presents the Test-Time adjustment by Voronoi Diagram (TTVD) framework by leveraging geometric principles, particularly the Voronoi Diagram (VD) and its extensions: the Cluster-induced Voronoi Diagram (CIVD) and the Power Diagram (PD). TTVD addresses the limitations of current test-time methods by using these geometric structures to improve feature alignment and sample filtering. CIVD enhances robustness by considering clusters rather than individual prototypes, while PD allows flexible boundaries to better handle noisy samples near decision boundaries. The proposed TTVD demonstrates substantial improvements over state-of-the-art TTA methods on several corrupted datasets, showing its effectiveness in real-world distribution shift scenarios.

**Strengths:**

- By introducing geometric frameworks like VD, CIVD, and PD, TTVD leverages computational geometry to improve the alignment of test-time features with training distributions. This approach provides a mathematically grounded and visually interpretable solution to feature adaptation in TTA.
- TTVD shows consistent improvements over existing methods across multiple datasets, reducing classification error rates and enhancing model calibration as indicated by lower Expected Calibration Error (ECE) scores. The inclusion of diverse corruption types in the evaluation (e.g., noise, blur, and weather-based distortions) demonstrates the framework’s adaptability to real-world conditions.

**Weaknesses:**

- Both CIVD and PD are well-established geometric structures, raising questions about the novelty of TTVD’s core contributions. The first two contributions mainly apply these established methods to the TTA setting, which may limit the originality of the approach.
- The distinction between “test-time training” (TTT) and “test-time adaptation” (TTA) is somewhat blurred. According to the TENT framework, TTA excludes access to source data, while TTT can include self-supervised losses on source data. TTVD’s reliance on pre-computed Voronoi sites calculated during pre-training suggests it should be categorized as TTT rather than TTA. This distinction impacts baseline comparisons, as the current baselines primarily include TTA methods, potentially leading to an unfair performance comparison.
- The paper claims that TTVD extends VD from a point-to-point structure to a cluster-to-point influence mechanism, but it’s unclear why distances in standard VD (calculated by $\mu_k$) would not already reflect cluster-to-point relationships. A clearer explanation of this transition’s significance would be beneficial.
- The method lacks details on integrating VD, CIVD, and PD into a single loss function. Questions remain regarding whether the label y is generated by substituting $d(\cdot)$ in Equation 3 with $F(\cdot)$, how the components are balanced, and whether this balance is sensitive to different datasets.
- The paper lacks computational details on estimating key parameters ($\mu, C, and~ v$) in the TTVD framework. More clarity on these calculations would enhance understanding of the implementation and reproducibility of TTVD.

**Questions:**

The reviewer may have limited familiarity with the Voronoi Diagram, which could have led to some misunderstandings. The authors are encouraged to provide additional explanations during the rebuttal to clarify above points, especially the contributions of the paper and the assumptions between TTT and TTA.

---

> ### Author Response · Authors · 2024-11-21
> **Response to Reviewer wGB6 (1/2)**
>
> We thank Reviewer wGB6 for the positive feedback and valuable review!
>
> **Response to Weakness 1:**
> We will answer your concern below regarding your Question 1.
>
> **Response to Weakness 2:**
> Thank you for pointing this out. We agree that the idea of "fully TTA" (carefully quoted here from TENT) excludes access to source data. However, as modern methods continue to evolve, it can be observed that SODA methods often do not strictly adhere to these settings. For instance, NOTE[1] requires pre-training from scratch to implement its proposed batch normalization layers, and AdaNPC[2] similarly relies on pre-training with its proposed KNN-based loss function. From their experiments, we observed that they primarily compare their results to the same baseline we used.
>
> While categorizing previous TTA methods is not the main focus of our paper, we think that their primary distinction lies in their methodology rather than their settings: TTT employs self-training, whereas TENT utilizes entropy minimization. Since our adaptation approach aligns more closely with entropy minimization, we included these baselines for comparison. However, we understand your concern and have added additional baselines from more recent methods that incorporate self-training or require modifications to the pre-training process.
>
> |                             | ImageNet-C |
> | --------------------------- | ---------- |
> | IST (CVPR 2024)[3]             | 63.4       |
> | MemBN (ECCV 2024)[4]           | 65.6      |
> | Decorruptor (ECCV 2024)[5]     | 63.8       |
> | TTVD (Ours)                 | 59.8       |
>
> **Response to Weakness 3:**
> In standard VD, each Voronoi cell is dominated/influenced by a single site $\mu_k$, as Eq. 2 shown. In contrast, in CIVD, each cell is influenced by a cluster of sites $\mathcal{C}_k$, as Eq. 4 shown. This transition to multi-site influence enhances the robustness of the cells. Essentially, a standard VD can be viewed as a special case of 1-nearest neighbor, as VD serves as a foundational structure for nearest-neighbor methods. We have added a figure at the end of Section 3 to illustrate the differences among VD, CIVD, and CIPD, providing a clearer explanation.
>
> **Response to Weakness 4:**
> You are right that label y is generated by replacing $d$ with $F$. However, there seems to be a misunderstanding regarding our proposed method. In fact, VD, CIVD, CIPD are seperate structures. We have revised our summary of them at the end of Section 3 for clarity. VD → CIVD → CIPD represent a progressive relationship rather than a combination. VD is the simplest point-to-point structure among the three. CIVD builds upon it as a cluster-to-point structure, while CIPD further extends CIVD by incorporating the power distance, as described in Eq. 5. We have added the pseudo-code for CIVD and CIPD in Appendix H. We use CIPD for all datasets, as it is the most advanced structure of the three.
>
> **Response to Weakness 5:**
> Thank you for your advice. We submitted our code in the supplementary to help public readers understand and reproduce our method. $\mu$ is calculated from the class means of the training set, which is stated under the ``implementation details'' in the experiment section. To generate $\mathcal{C}$, we use self-supervision to expand the Voronoi sites, which is stated in detail in Section 3.2. For example, we use rotation to expand $K$ sites to $4K$ sites, and every 4 sites form a cluster $\mathcal{C_k}$. $v$ is set according to Lemma 3.1, where it can be calculated from the classifier layer of the pretrained model.
>
> **Response to Question 1:**
> We are grateful for the opportunity to explain! We understand that this paper is intensive on Voronoi Diagrams, a classical structure from computational geometry that are not widely explored in the field of machine learning, which may challenge the readability.
>
> As reviewer ET99 commented, our method is the first to apply VD for Test-time Adaptation, even though VDs were originally developed for space partition. Our contribution is mainly about the exploration between VDs, neighbor-based methods and TTA. First, we revealed that nearest neighbor algorithms can be analyzed through standard VD. Then, we find that advanced VDs, such as CIVD, PD, offer benefits for TTA, because of their unique properties (enhanced robustness from multi-site influence, more flexible partitions). **We would like to note that previous papers on CIVD mainly focus on theoretical aspects in geometry [6][7][8]. In contrast, our work seeks to explore its potential, bring renewed attention to it, and transition it into machine learning applications, specifically TTA.**

---

> > ### Author Response · Authors · 2024-11-21
> > **Response to Reviewer wGB6 (2/2)**
> >
> > [1] Taesik Gong, et al. NOTE:Robust continual test-time adaptation against temporal correlation. In Advances in Neural Infor-
> > mation Processing Systems, 2022.
> >
> > [2] Zhang, Yifan, et al. "Adanpc: Exploring non-parametric classifier for test-time adaptation." International Conference on Machine Learning. PMLR, 2023.
> >
> > [3] Ma, Jing. "Improved Self-Training for Test-Time Adaptation." Proceedings of the IEEE/CVF Conference on Computer Vision and Pattern Recognition. 2024.
> >
> > [4] Kang, Juwon, et al. "MemBN: Robust Test-Time Adaptation via Batch Norm with Statistics Memory." European Conference on Computer Vision. Springer, Cham, 2025.
> >
> > [5] Oh, Yeongtak, et al. "Efficient Diffusion-Driven Corruption Editor for Test-Time Adaptation." arXiv preprint arXiv:2403.10911 (2024).
> >
> > [6] Danny Z. Chen, et al. On clustering induced voronoi diagrams. In 2013 IEEE 54th Annual Symposium on Foundations of Computer Science.
> >
> > [7] Danny Z. Chen, et al. On clustering induced voronoi diagrams. SIAM Journal on Computing, 46(6):1679–1711, 2017.
> >
> > [8] Ziyun Huang, et al. Influence-based voronoi diagrams of clusters. Computational Geometry, 96:101746, 2021a. ISSN 0925-7721.

---

> > > ### Author Response · Authors · 2024-11-25
> > >
> > > Thank you once again for your time and effort providing the initial review of our paper! We have carefully addressed the questions you raised and provided detailed responses above. As the discussion period nears its end, we kindly invite you to share any feedback you might have, particularly regarding the geometric perspective of our contributions, and we would be happy to discuss them further.

---

> > > > ### Comment · Reviewer_wGB6 · 2024-11-25
> > > > **response**
> > > >
> > > > The rebuttal provides the differences among three types of VD and experiments with TTT methods. I think my most concerns have been addressed.

---

> > > > > ### Author Response · Authors · 2024-11-25
> > > > >
> > > > > We sincerely thank you for raising the score!

---

### Official Review · Reviewer_rD6K · 2024-10-31

**Soundness:** 2
**Presentation:** 2
**Contribution:** 2
**Rating:** 6
**Confidence:** 4

**Summary:**

The paper targets the challenge of test-time adaptation (TTA) in deep learning models. The authors propose a framework, TTVD (Test-Time adjustment by Voronoi Diagram guidance), which leverages the geometric properties of Voronoi Diagrams to adapt models online during inference. The paper introduces two key geometric structures: Cluster-induced Voronoi Diagram (CIVD) and Power Diagram (PD), to enhance the robustness and adaptability of models facing distributional shifts. Extensive experiments on benchmark datasets like CIFAR-10-C, CIFAR-100-C, ImageNet-C, and ImageNet-R demonstrate the effectiveness of TTVD against state-of-the-art methods.

**Strengths:**

1. The paper offers a fresh perspective on TTA by employing Voronoi Diagrams, which is a significant departure from traditional approaches and shows promise in handling distributional shifts.
2. The authors provide a rigorous experimental evaluation, demonstrating TTVD's superiority over existing methods on multiple benchmark datasets, which strengthens the credibility of their approach.
3. Leveraging Voronoi Diagrams for TTA enhances model interpretability, allowing for clearer visualizations and understanding of partition boundaries, which is a valuable asset in deep learning.

**Weaknesses:**

1. While the paper discusses the benefits of TTVD, it lacks a detailed discussion on the computational overhead introduced by the geometric structures, which could be a concern for real-time applications.
2. While experimental results are promising, it would be valuable to see a comparison with theoretical bounds or guarantees, if available, to understand the limits of TTVD.
3. Some sections, particularly the methodology, could benefit from more detailed explanations or pseudo-code to aid reproducibility.
4. The performance of geometric structures like CIVD and PD may be sensitive to hyperparameters. The paper could provide more insights into hyperparameter tuning and the robustness of these parameters.

**Questions:**

See weaknesses.

---

> ### Author Response · Authors · 2024-11-21
> **Response to Reviewer rD6K**
>
> Thank you for your constructive comments. We address the concerns as follows.
>
> **Response to Weakness 1:**
> We report the GPU memory usage and process time for each batch on CIFAR-C as below. The memory usage is similar to TTT, while TTVD is faster and similar to TENT.
>
> |         | Memory Usage (MiB) | Infer and adapt (ms) |
> | --------| -------- | --- |
> | Tent    | 614      | 31|
> | TTT             | 914      | 54|
> | TTVD            | 932      | 39|
>
> **Response to Weakness 2:**
> Your are right that a theoretical analysis of VDs would greatly benefit the understanding of our method. Actually, this is also a challenge for most of the current TTA methods. We will leave this a valuable future work.
>
> **Response to Weakness 3:**
> Thank you for your advice. We have added the pseudo-code for CIVD and CIPD in the Appendix H. We also included our code in the supplementary materials for public readers to understand and reproduce our method.
>
> **Response to Weekness 4:**
> The only hyperparameter introduced in our method is $\gamma$ that control the magnitude of influence. We find it does not affect the model's accuracy. We set $\gamma=-0.8$ as a heuristic value to appropriately scale the influence of distant sites. Additionally, we provide a toy 3D illustration of the influence function to help your understanding, showing that as the distance increases, the influence of a site diminishes (https://anonymous.4open.science/r/iclr-654F/output.png).
>
> | gamma           | Error (ImageNet-C)     | ECE (ImageNet-C) |
> | --------------- | --------- | --- |
> | 0.9             | 59.8      | 17.8|
> | 0.8             | 59.8      | 21.0|
> | 0.7             | 59.7      | 23.3|

---

> > ### Author Response · Authors · 2024-11-25
> >
> > Thank you once again for your time and effort providing the initial review of our paper! We have carefully addressed the questions you raised and provided detailed responses above. As the discussion period nears its end, we kindly invite you to share any feedback you might have, and we would be happy to discuss them further.

---

### Official Review · Reviewer_ET99 · 2024-11-04

**Soundness:** 3
**Presentation:** 2
**Contribution:** 3
**Rating:** 6
**Confidence:** 4

**Summary:**

This paper uses a combination of different variants of Voronoi Diagrams in test-time adaptation. The proposed method combines original Voronoi Diagram, Cluster-induced Voronoi Diagram, and Power Diagram. The proposed method outperforms a collection of relevant baselines under 4 benchmarking datasets.

**Strengths:**

- The proposed method is novel, seems to be the first working applying Voronoi Diagram in test-time adaptation, although there are works (e.g. T3A, AdaNPC) share similar intuitions.
- The author choose very appropriate baselines: all of them are highly relevant and shares similarities to the proposed method. The proposed method has strong performance.

**Weaknesses:**

- The algorithm part seems unfinished and lacks many details. For example, the paper only include how to compute the soft prediction $\hat{y}$ for VD, but not for CIVD and PD. Also, it is not introduced how these three components are combined and whether there are additional hyper parameters or flexibilities.
- It seems like this paper changes the way of doing inference (from simple linear layer to a combination of three types of Voronoi Diagrams). However, the TTA process is still just entropy minimization, like a simpler version of SHOT. Given this similarity, it is highly unsure why the proposed method can solve the challenges in introduction, and how.
- [Minor] The format of references may need to be updated. There are many places where the author use \cite, while it should be \citep. Please correct it in the next version.

**Questions:**

- For the Voronoi Diagram method in Section 3.1, Is it true that the Voronoi sites won’t be updated once initialized? Since in Algorithm 1, it is not adapted.
- How to get the soft prediction based on $F$ in formula (4) and (6)? How three diagrams are combined?
- What is the purpose of Lemma 3.1? Is it how the $v_k$s are initialized?

---

> ### Author Response · Authors · 2024-11-21
> **Response to Reviewer Reviewer ET99**
>
> We appreciate your time and effort in reviewing our manuscript. Here are the clarification and answers to the concerns.
>
> **Response to Weakness 1:**
> Thank you for your advice. We have added the pseudo-code for CIVD and CIPD. However, there seems to be a misunderstanding regarding our proposed method, possibly due to our wording. In fact, VD, CIVD, CIPD are seperate structures. We have revised our summary of them at the end of Section 3 for clarity. VD → CIVD → CIPD represent a progressive relationship rather than a combination. VD is the simplest point-to-point structure among the three. CIVD builds upon it as a cluster-to-point structure, while CIPD further extends CIVD by incorporating the power distance, as described in Eq. 5. Our experiments demonstrate that their performance follows the order: CIPD > CIVD > VD.
>
> As defined in Eq. 4 and Eq. 6, only a single hyperparameter, $\gamma$, is introduced in both CIVD and CIPD. We find that $\gamma$ does not affect the accuracy performance a lot, and we set $\gamma=-0.8$ as a heuristic value to appropriately scale the influence of distant sites.
>
> **Response to Weakness 2:**
> Your are right that we do inference using VDs instead of linear layers. Actually, not only SHOT but also many current TTA methods (TENT, NOTE, Conjugate PL, SAR) utilize entropy minimization. The unique properties of CIVD and CIPD contributing to the overall improvement. CIVD introduces multi-site influences, enhancing robustness, while CIPD enables more flexible partitioning by incorporating the power distance function.
>
> **Response to Minor Weakness 3:**
> Thank you for your advice. We have revised them in our manuscript.
>
> **Response to Question 1:**
> Yes, the Voronoi sites won't be updated. They are served as the adaptation guidance.
>
> **Response to Question 2:**
> To get the soft prediction, simply replace $d$ with $F$ in Equation 3.
>
> **Response to Question 3:**
> $v$ is set according to Lemma 3.1, where it can be calculated from the classifier layer of the pretrained model.

---

> > ### Author Response · Authors · 2024-11-25
> >
> > Thank you once again for your time and effort providing the initial review of our paper! We have carefully addressed the questions you raised and provided detailed responses above. As the discussion period nears its end, we kindly invite you to share any feedback you might have, and we would be happy to discuss them further.

---

> > > ### Comment · Reviewer_ET99 · 2024-12-02
> > > **Thanks for your rebuttal!**
> > >
> > > Thanks for your rebuttal. I get your point that VD → CIVD → CIPD represent a progressive relationship rather than a combination. The revision of the paper makes it much more clear (e.g. line 354 - 357). Most of my concerns are addressed. The new diagram also helps understanding, thanks for your efforts! **I raised my rating to 6, good luck!**
> > >
> > > I am still curious about the synergy between entropy minimization and the proposed CIPD. In my opinion, CIPD mostly changes the "inference" phase, this makes me wondering:
> > > 1. If we do not update the pretrained feature extractor, how much performance gain we can get, compared to the full algorithm? This can be tested by setting the learning rate to be zero (if you don't have running mean / running variance or use the model.eval mode)
> > > 2. Since the feature extractor is updated, but the Voronoi sites won't be updated, I feel this part to be counter-intuitive: It seems like either (1) the feature extractor won't be significantly adapted, or (2) the Voronoi sites in the beginning and the end of testing sequence are not guaranteed to match (or drawn from the same distribution).
> > > 3. Since entropy is known to have trivial solution (e.g. Figure 3(a) in [1]), do you observe any model collapse when the learning rate is too high? Like in the testing sequence, the model performance first increase but then decrease. Is it possible that CIPD can help alleviating the trivial solution problem?
> > >
> > > [1] Hao Zhao, Yuejiang Liu, Alexandre Alahi, Tao Lin. On Pitfalls of Test-Time Adaptation. ICML 2023.
> > >
> > > I really apologize to ask questions at the end of the discussion phase. And I think some of these questions can be out of the scope o this paper. So you don't have to feel obliged to answer these question in just one day. But I think these questions can help better understanding the synergy between inference and adaptation.

---

> > > > ### Author Response · Authors · 2024-12-03
> > > >
> > > > We sincerely thank you for raising the score! Your feedback is valuable and constructive in helping us improve our paper. Due to time limit, we feel sorry that we are unable to provide results for the additional experiments on the final day of the discussion period, but we will certainly discuss them in the next version of our paper!

---

### Official Review · Reviewer_WP62 · 2024-11-04

**Soundness:** 4
**Presentation:** 4
**Contribution:** 3
**Rating:** 6
**Confidence:** 2

**Summary:**

This paper introduces a novel Test-Time Adaptation (TTA) method using Voronoi Diagrams, termed TTVD. The manuscript highlights the integration of cluster-induced Voronoi Diagrams with Power Diagrams, marking their inaugural application in the TTA domain. This combination aims to ensure both flexibility and robustness within the method. Based on the experiments presented, TTVD demonstrates a clear reduction in errors, which is commendable. The writing is clear, the methodology sound, and the experimental outcomes show significant improvement. However, the manuscript does raise concerns about its level of innovation, primarily since it builds upon pre-existing methodologies without introducing novel concepts.

**Strengths:**

1. The proposed TTVD method is presented as both simple and rational, making it easily understandable.
2. The experimental performance of the TTVD method is impressive and showcases the method’s efficacy.

**Weaknesses:**

1. The main innovation of TTVD appears to be the combination of existing methods (Cluster-induced Voronoi Diagram and Power Diagram), which might not sufficiently fulfill the criteria for substantial novelty.
2. The manuscript lacks a comprehensive discussion and validation of parameter settings, such as \gamma, \eps, and \tau, which are crucial for the reproducibility and understanding of the research.
3. The comparative analysis primarily focuses on methods proposed up to and including 2023. Given the rapid advancements in the field, incorporating more recent methodologies (from 2024) could provide a more current understanding of TTVD's positioning.

**Questions:**

1. The parameter gamma seems to be a critical aspect of TTVD; however, its determination and impact on algorithm performance are not thoroughly discussed in the manuscript. Could you provide a detailed explanation on how gamma values are selected and their influence on the method's efficacy?
2. Regarding the introduction of parameter eps in equation (3) to avoid the log0 issue, the justification seems unclear. The rationale that including eps prevents log0 errors in equation (3) is not compelling, as the log0 problem might not arise even without eps. Could you elaborate on the necessity of eps in this context?
3. The manuscript assumes the Voronoi sites are pre-determined without detailing the process of converting Xtest into Voronoi sites. For clarity and completeness, please provide a comprehensive description of how Xtest data are transformed into Voronoi sites.

---

> ### Author Response · Authors · 2024-11-21
> **Response to Reviewer WP62**
>
> We sincerely thank Reviewer WP62 for the valuable time and effort in reviewing our work.  We address the concerns as follows.
>
> **Response to Weakness 1:** As reviewer ET99 commented, our method is the first to apply VD for Test-time Adaptation, even though VDs were originally developed for space partition. Our contribution is mainly about the exploration between VDs, neighbor-based methods and TTA. First, we revealed that nearest neighbor algorithms can be analyzed through standard VD. Then, we find that advanced VDs, such as CIVD, PD, offer benefits for TTA, because of their unique properties (enhanced robustness from multi-site influence, more flexible partitions). **We would like to note that previous papers on CIVD mainly focus on theoretical aspects in geometry[1][2][3].  In contrast, our work seeks to explore its potential, bring renewed attention to it, and transition it into machine learning applications, specifically TTA.**
>
> **Response to Weakness 2:** We explain the setting of $\gamma$, $\epsilon$, and $\tau$ as follows.
>
> $\epsilon$ is the machine epsilon, i.e. the smallest number that a computer can recognize as being greater than zero, but still very small in magnitude. It is used for code implementation, to improve the numerical stability, rather than a hyperparameter for the algorithm. We set it as 1e-8, similar to the epsilon used in Pytorch Adam implementation.
>
> $\tau$ is temperature of the softmax function. We use the standard softmax function, where $\tau = 1$. Both $\epsilon$ and $\tau$ are included in the equation for completeness and are set according to common implementation practices, rather than being treated as hyperparameters to tune in our framework.
>
>
> $\gamma$ is the parameter to control the magnitude of the influence. We use a small negative value to scale and diminish the influence of distant sites. For example, we provide a toy 3D illustration of the influence function to show that as the distance increases, the influence of a site becomes smaller (https://anonymous.4open.science/r/iclr-654F/output.png). We find that $\gamma$ does not affect the accuracy performance a lot from below, and we set $\gamma=-0.8$ as a heuristic value to appropriately scale the influence of distant sites.
>
> | gamma           | Error (ImageNet-C)    | ECE (ImageNet-C) |
> | --------------- | --------- | --- |
> | 0.9             | 59.8      | 17.8|
> | 0.8             | 59.8      | 21.0|
> | 0.7             | 59.7      | 23.3|
>
> Aside from these, common parameters such as the learning rate are configured according to the TTAB framework, as discussed in Appendix D. No additional parameters are introduced in our geometric framework.
>
> **Response to Weakness 3:**
> We included 3 more recent methods to compare as below. It is worth noting that, since our method is based on VDs, it is orthogonal to most current approaches. This means it can be combined with other methods to further enhance performance.
>
> |                             | Error (ImageNet-C) |
> | ----------------- | ---------- |
> | IST (CVPR 2024)[4]       | 63.4    |
> | MemBN (ECCV 2024)[5]           | 65.6      |
> | Decorruptor (ECCV 2024)[6]     | 63.8       |
> | TTVD (Ours)                 | 59.8       |
>
> **Response to Question 1:**
> We explained this in Weakness 2 above and it does not affect the model's accuracy.
>
> **Response to Question 2:**
> Epsilon is a very small value used to enhance numerical stability in the code implementation, particularly when calling the log-softmax function in PyTorch. This is similar to the Adam optimizer implementation in PyTorch. We agree that the log problem might not arise, but we include this detail in the manuscript for completeness, and it does not affect the performance.
>
> **Response to Question 3:**
> Thank you for your advice. Voronoi sites are computed from the training set and the adaptation follows them using test data. We stated in the experiment section, under the "implementation details".Additionally, we have included a figure at the end of Section 3 to illustrate the workflow of our method. Furthermore, we shown that our method is robust to the precision of the Vonronoi sites in Table 4.
>
> [1] Danny Z. Chen, et al. On clustering induced voronoi diagrams. In 2013 IEEE 54th Annual Symposium on Foundations of Computer Science.
>
> [2] Danny Z. Chen, et al. On clustering induced voronoi diagrams. SIAM Journal on Computing, 46(6):1679–1711, 2017.
>
> [3] Ziyun Huang, et al. Influence-based voronoi diagrams of clusters. Computational Geometry, 96:101746, 2021a. ISSN 0925-7721.
>
> [4] Ma, Jing. "Improved Self-Training for Test-Time Adaptation." Proceedings of the IEEE/CVF Conference on Computer Vision and Pattern Recognition. 2024.
>
> [5] Kang, Juwon, et al. "MemBN: Robust Test-Time Adaptation via Batch Norm with Statistics Memory." European Conference on Computer Vision. Springer, Cham, 2025.
>
> [6] Oh, Yeongtak, et al. "Efficient Diffusion-Driven Corruption Editor for Test-Time Adaptation." arXiv preprint arXiv:2403.10911 (2024).

---

> > ### Author Response · Authors · 2024-11-25
> >
> > Thank you once again for your time and effort providing the initial review of our paper! We have carefully addressed the questions you raised and provided detailed responses above. As the discussion period nears its end, we kindly invite you to share any feedback you might have, and we would be happy to discuss them further.

---

> > ### Comment · Reviewer_WP62 · 2024-11-26
> >
> > Thank you for your response, which has addressed some of my concerns. As a result, I have slightly increased my score. However, overall, I remain somewhat dissatisfied with the level of innovation in the proposed approach, particularly in terms of the insights and impact it offers to researchers in the field.

---

> > > ### Author Response · Authors · 2024-12-03
> > >
> > > We sincerely thank you for raising the score!

---

### Author Response · Authors · 2024-11-21
**Summary of the rebuttal revisions.**

We sincerely thank Reviewers WP62, ET99, rD6K, and wGB6 for their constructive comments. Below, we summarize the changes made in the revision, which are highlighted in red text.

1. **Additional Figure to Explaining VD, CIVD and CIPD** (ET99, wGB6):
We have revised the summary of VD, CIVD, and CIPD at the end of Section 3 (Methodology) and added Figure 3 to illustrate their differences, clarifying the overall pipeline of our method. The original wording may have caused some misunderstanding of our method. In fact, VD → CIVD → CIPD represent a progressive relationship rather than a combination. Our proposed TTVD is constructed progressively, transitioning from standard VD to CIVD and, finally, to CIPD.

2. **Algorithms for CIVD and CIPD** (ET99, rD6K):
We have added the pseudo-code for CIVD and CIPD. In fact, CIVD and CIPD are advanced forms of VDs, which can be implemented by simply replacing $d$ with $F$ in Algorithm 1. Please note that we have also included our code in the supplementary materials to help public readers better understand our method.

Here are some common questions that reviewers raise regarding our work.

1. **Novelty** (WP62, wGB6): As Reviewer ET99 commented, our method is the first to apply VD for Test-time Adaptation, even though VDs were originally developed for space partition. Our contribution is mainly about the exploration between VDs, neighbor-based methods and TTA. First, we revealed that nearest neighbor algorithms can be analyzed through standard VD. Then, we find that advanced VDs, such as CIVD, PD, offer benefits for TTA, because of their unique properties (enhanced robustness from multi-site influence, more flexible partitions). We would like to note that previous papers on CIVD mainly focus on **theoretical aspects in computational geometry[1][2][3]**. In contrast, our work seeks to **explore its potential, bring renewed attention to it, and transition it into machine learning applications, specifically TTA.**

2. **Hyperparameters** (WP62, ET99, rD6K, wGB6): The only hyperparameter introduced in our method is $\gamma$ that control the magnitude of influence. We found that it does not affect the model's accuracy. Other parameters such as $\epsilon$ and $\tau$ are included in the paper for completeness and are set following common practice (e.g., using the standard softmax function). As stated in Section 4 (Experiment), Voronoi sites are precomputed from the dataset. We have added this explanation in Appendix D.

Thank you all for your comments and suggestions! Please kindly let us know if you have any further concerns or feedback.

[1] Danny Z. Chen, et al. On clustering induced voronoi diagrams. In 2013 IEEE 54th Annual Symposium on Foundations of Computer Science.

[2] Danny Z. Chen, et al. On clustering induced voronoi diagrams. SIAM Journal on Computing, 46(6):1679–1711, 2017.

[3] Ziyun Huang, et al. Influence-based voronoi diagrams of clusters. Computational Geometry, 96:101746, 2021a. ISSN 0925-7721.

---

### Meta-Review · Area_Chair_5afa · 2024-12-18

**Metareview:**

Thanks for your submission to ICLR.  This paper received four reviews, and the reviewers ultimately agreed that the paper is sufficiently strong for publication.  On the positive side, reviewers noted the novelty of the method, and its good empirical results.  On the negative side, some reviewers felt that the method was incremental, and several reviewers noted some details missing from the paper throughout.

During the discussion period, the author rebuttal addressed many of the reviewer concerns, and several of the reviewers raised their scores.  At this point, all four reviewers are leaning accept on this paper, and I am happy to recommend accepting the paper for publication.

Please do try to address the reviewer comments in the final version of the paper.

**Additional Comments On Reviewer Discussion:**

Three of the four reviewers raised their score during the discussion.  One reviewer (who already had a positive score overall) did not participate in the discussion.  Overall, it seems that the rebuttal helped to clear up major concerns about the paper.

---

### Decision · Program_Chairs · 2025-01-22

Accept (Poster)